# Differential effect of supercoiling on bacterial transcription in topological domains

Boaz Goldberg[1], Nicolás Yehya[1], Jie Xiao[1*], Sam Meyer[1,2*]

**1** Department of Biophysics and Biophysical Chemistry, Johns Hopkins School of Medicine, Baltimore, Maryland, United States of America, **2** Université de Lyon, INSA Lyon, Université Claude Bernard Lyon 1, CNRS UMR5240, Laboratoire de Microbiologie, Adaptation et Pathogénie, Villeurbanne, France

\* xiao@jhmi.edu (JX); sam.meyer@insa-lyon.fr (SM)

## Abstract

DNA supercoiling (SC), the over- and under-winding of DNA, is generated by transcription as described in the twin-domain model. Conversely, SC also impacts transcription through torsional stress. SC therefore regulates transcription dynamically and independently of transcription factor binding, particularly in the context of chromosomal topological domains and the activity of topoisomerases in bacteria. In this work, we develop numerical simulations of SC-coupled transcription of a single gene within a topological domain, based on a model incorporating stochastic transcription and activities of topoisomerase I and gyrase. We explore the effect of several parameters not systematically assessed in previous works (role of topoisomerase activities, topological domain size, gene expression strength) and compare the simulation results to a diverse set of experimental observations ranging from *in vitro* transcription assays to transcriptomics datasets from various species. This model recapitulates the non-monotonic dependence of transcription *in vitro* with the superhelical density of the plasmid template. Simulations of *in vivo* transcription in a closed domain exhibit a qualitatively different role for the two topoisomerases, as well as qualitatively different regulatory behaviors depending on the promoter strength. Specifically, topoisomerase I is required for strongly expressed genes that may be hindered by stalled RNA Polymerase, whereas gyrase activity favors the expression of all genes by enhancing transcription initiation and modulating the burstiness of transcription. The simulations exhibit a new mechanism for transcription bursting mediated by negative SC accumulating at the promoter region and modulating the initiation rate, resulting in levels of burstiness compatible with values reported in cells. Finally, we analyze several transcriptomics datasets from a range of evolutionarily distant species and show that topoisomerase inhibition is systematically associated with the repression of highly expressed genes. Simulations show this behavior to occur within a limited parameter range and thus indicate a biologically relevant regime for the simulations. Overall, this work provides a more quantitative description of how SC contributes to differential gene regulation and transcriptional bursting in bacteria.

**Data availability statement:** All codes and analysis scripts are available from the Zenodo database at DOI: https://doi.org/10.5281/zenodo.14586118.

**Funding:** This study was supported by the Johns Hopkins University Provost's Undergraduate Research Award 2024 to B. G., the National Science Foundation MCB1817551 and MCB2412916 to J. X., the Agence Nationale de la Recherche ANR-24-CE95-0003-01 to S.M. The funders played no role in study design, data collection and analysis, decision to publish, or preparation of the manuscript.

**Competing interests:** The authors have declared that no competing interests exist.

## Author summary

We are interested in understanding how bacteria regulate the expression of their genes independent of protein-based transcription factors. One mechanism is through DNA mechanics caused by the over- and under-winding of the DNA, termed DNA supercoiling. Because bacteria can use enzymes called topoisomerases to regulate the supercoiling level of the DNA, these enzymes can serve as gene regulators. It is difficult, however, to understand how topoisomerases act as gene regulators through experiments alone. Currently, there is no method of measuring the supercoiling level along a stretch of DNA over an extended period in living cells. We therefore developed a computational model of gene transcription that accounts for supercoiling dynamics. Unlike some previously existing models, we consider continuous response curves of topoisomerases' activities in response to the local supercoiling level and the ability of gyrase to perform several catalytic cycles per binding event. We also perform extensive comparisons between our model and existing gene expression and transcriptomics data, which is essential to ensuring the biological relevance of computational modeling efforts. We replicate and provide detailed explanations for several experimental observations, including the connection between supercoiling and transcriptional bursts and the selective gene expression modulation by topoisomerase inhibition that is based on the promoter strength.

## Introduction

DNA supercoiling (SC), the over- (positive) and under-winding (negative) of double-stranded DNA (dsDNA) relative to its relaxed B-form state, has been proposed as a means to regulate transcription independent of regulatory protein binding [1]. Negative SC facilitates transcription initiation by driving promoter melting during the formation of the open complex consisting of DNA and RNA polymerase (RNAP) [2]. During transcription elongation, SC-induced torque can stall RNAP [3]. Conversely, transcription also influences SC: an elongating RNAP molecule introduces negative SC upstream and positive SC downstream per the twin-domain model due to the compensatory DNA rotation caused by the rotational drag of the transcription [4,5]. Moreover, SC can diffuse over distances of several kilobases, allowing for cooperative and antagonistic interaction of multiple RNAP molecules within and between genes [6]. Diffusion of SC can be blocked by topological barriers formed by protein binding and/or protein-induced DNA loops, which prevent dsDNA rotation [7,8] and topologically isolate genes from those on the other side of the barriers. Given the coupled interactions between SC and transcription, SC in topological domains may serve as a potent and complex transcriptional regulator independent of protein transcription factors, leading to a DNA-based transcription regulatory network to act upon adjacent genes across long distances.

Cells regulate the average SC level of their chromosomes using topoisomerases. In bacteria, topoisomerase I (TopoI) and gyrase [9] maintain SC homeostasis. TopoI

is a type I topoisomerase that relaxes negative SC by introducing transient single-stranded breaks in dsDNA in an ATP-independent manner; gyrase is a type II topoisomerase that hydrolyzes ATP to relax positive SC and introduce negative SC by passing one dsDNA segment through a transient double-stranded break of another [10–12]. Given the interactions of SC and transcription, it is reasonable to expect that variations in topoisomerases with antibiotics alter transcription and that this alteration varies between genes in a manner dependent on growth phase, expression strength, and genomic context [10,13–22].

While SC may serve as a new, important transcriptional regulator, our understanding is limited by experimental difficulties in measuring the spatiotemporal dynamics of SC along the chromosome and its impact on transcription kinetics. Computational modeling can explore various aspects of SC and its coupling with transcription, helping identify possible underlying mechanisms and providing testable hypotheses for further experimental development [23–25]. In the past years, we and others have developed multiple models of coupled dynamics between bacterial transcription and SC, either with a mathematical framework [23,24,26,27] or with a computational biophysics focus [16,21,25,28–31]. Some models considered the regulatory effect of SC on transcription initiation [16,32], elongation [30,33], or both [21,25,27–31]. Some explicitly described the activities of both gyrase and TopoI [16,21,27,28,31], or the bursting properties of transcription [34]. Additional models relevant to eukaryotic transcription also exist [25,35].

While these past studies significantly improved our understanding of the relationship between SC and transcription, only a few of them [16,21,27–29,34] quantitatively compared modeling results with experimental data, primarily focusing on that of Kim et al. [6]. Nevertheless, numerous other transcription datasets have been collected since the 1980s, providing valuable, diverse, and complementary quantitative information on the role of topoisomerases and topological constraints on gene expression, including in their native context. These data range from *in vitro* transcription studies involving plasmids prepared at different superhelical densities [36–38], to whole-genome transcriptomics data observed under varying superhelical densities induced by drugs or topoisomerase [13–20,36–39], as well as studies of regulatory interactions between neighboring genes in various orientations, based on artificial constructions on the chromosome or plasmids [24,27,40,41]. Systematic integration and interpretation of these data with computational modeling is therefore an important task in the field. To the best of our knowledge, only one computational study has incorporated transcriptomics data in its analysis [16], which only included two bacterial species (*Escherichia coli* and *Dickeya dadantii*). In this study, we aim to integrate quantitative simulations of transcription with diverse experimental data, including a systematic analysis of transcriptomes from many species.

While previous modeling works have considered several adjacent genes [16,27–30,32,33], the observed complexity of these models motivated us to simulate a single, isolated gene, under varying topological conditions. Where experimental phenomena can be reproduced in a single gene simulation, the reduced complexity allows us to extract the clearest proposed mechanisms for these observations. A major difficulty in the field is that, due to the highly nonlinear and dynamical features of the process, quantitative variations in the parameters can lead to qualitatively diverging behaviors. Therefore, after presenting the new proposed model (see similarities and differences with previous models in Methods, and a review of the prior models in S1 Table), we systematically analyze the effect of three physiologically relevant quantitative parameters—topoisomerase activities, topological domain size, and promoter strength—on the transcription of a single gene and compare it to previously published data of direct relevance (*in vitro* and *in vivo*).

As we describe below, this modeling approach produced a remarkably rich and non-intuitive landscape of regulatory behaviors, in particular revealing very different effects for topoisomerase I and gyrase, and a qualitatively different response of weak and strong promoters to topological constraints. Our simulations provide a mechanistic basis for quantitative experimental observations, including: (1) the characteristic response curve of many promoters to varying SC *in vitro*, with optimal expression observed around physiological negative levels [36–38]; (2) the complex, promoter-dependent regulatory effect of varying topoisomerase activity *in vivo*, as well as topological domain sizes [13–21,39]; and (3) the emergence of transcriptional bursting in the absence of transcription factor binding [34]. Finally, we analyze a range of

transcriptomic datasets exhibiting the response of evolutionarily distant bacterial species to gyrase or topoisomerase I inhibition and show that the treatments systematically repress strongly expressed genes, albeit with different magnitudes. Simulations are compatible with the observed behavior in a part of the parameter space, indicating a biologically relevant regime for our simulations.

## Methods

### Modeling method overview

We simulate a one-dimensional topological domain containing a single gene of length $L$, with two flanking regions of distance $D$ to the domain barriers, giving an overall domain size of $2D+L$ (Fig 1). Our model is based on a modeling approach conceptually similar to several previous works, including ours [16,21,28–30], with differences and improvements summarized in S1 Table and in the dedicated paragraph below. The main elements of the model are described in Table 1 and are described in detail in the following paragraphs. At each timestep (of 0.04 s), transcription initiation may occur following a one-step SC-dependent stochastic initiation rate. Elongating RNAPs move by one bp, with a stalling threshold on upstream and downstream SC, upon reaching the terminator. Elongating RNAPs affect the SC levels upstream and downstream, which diffuse instantaneously (at the timescale of transcription and over kb-scale distances) and are modulated by the stochastic activities of topoisomerases.

### Computation of SC values

On a stretch of DNA, SC is quantified by the difference between the linking number—the number of times the DNA strands wind around each other—of the DNA stretch and that of an equivalent length of relaxed B-form dsDNA. The linking number consolidates both the twist (the wrapping of DNA strands around their helical axis) and writhe (the wrapping of the DNA duplex along itself) into a single number; as both configurations present similar topological constraints to RNAP, and supercoiling density is only related to the change in the linking number, we do not consider the distinction between these two forms of SC.

We consider a list of DNA-bound RNAP molecules ($RNAP_1$, …, $RNAP_k$), with $RNAP_1$ being furthest upstream (or the newest RNAP molecule on the DNA). Each RNAP is considered to occupy a DNA stretch of length $w=30$ bp to which other RNAP molecules cannot bind. For each $RNAP_i$, we track the location of its center, $x_i$, and the linking number of the DNA segment separating it from the nearest upstream $RNAP_{i-1}$ or barrier, $Lk_i^{(up)}$, and the nearest downstream $RNAP_{i+1}$ or barrier, $Lk_i^{(down)}$. From these linking numbers, we calculate a supercoiling density upstream ($\sigma_i^{(up)}$) and downstream ($\sigma_i^{(down)}$) of each RNAP molecule. The supercoiling density on a stretch of DNA is then defined by the formula

$$\sigma = \frac{Lk - Lk_0}{Lk_0}$$

where $Lk$ is the linking number of a DNA stretch and $Lk_0$ is the linking number of the most relaxed state of the same length of DNA (10.5 bp$^{-1}$). To manage the case where no RNAPs are bound, we also track $Lk_{dom}$, the linking number of the domain. We initialize $Lk_{dom}$ based on the initial supercoiling density, $\sigma_{start}$, an arbitrary value from which the simulation equilibrates. In the absence of topoisomerases, $Lk_{dom}$ is a constant value. Our simulation utilizes time steps of length $\Delta t = 1/v$, where $v = 25$ bp/s is the RNAP velocity [3,42].

### Initiation

We consider transcription initiation to occur in a single step facilitated by negative SC values. We assume that the concentration of RNAP is in great excess of DNA, so the concentration of the unbound RNAP pool stays constant. When initiation occurs, a new RNAP molecule is created with its center at the promoter position $D - \frac{w}{2}$, where $w$ is the footprint of an

PLOS Computational Biology

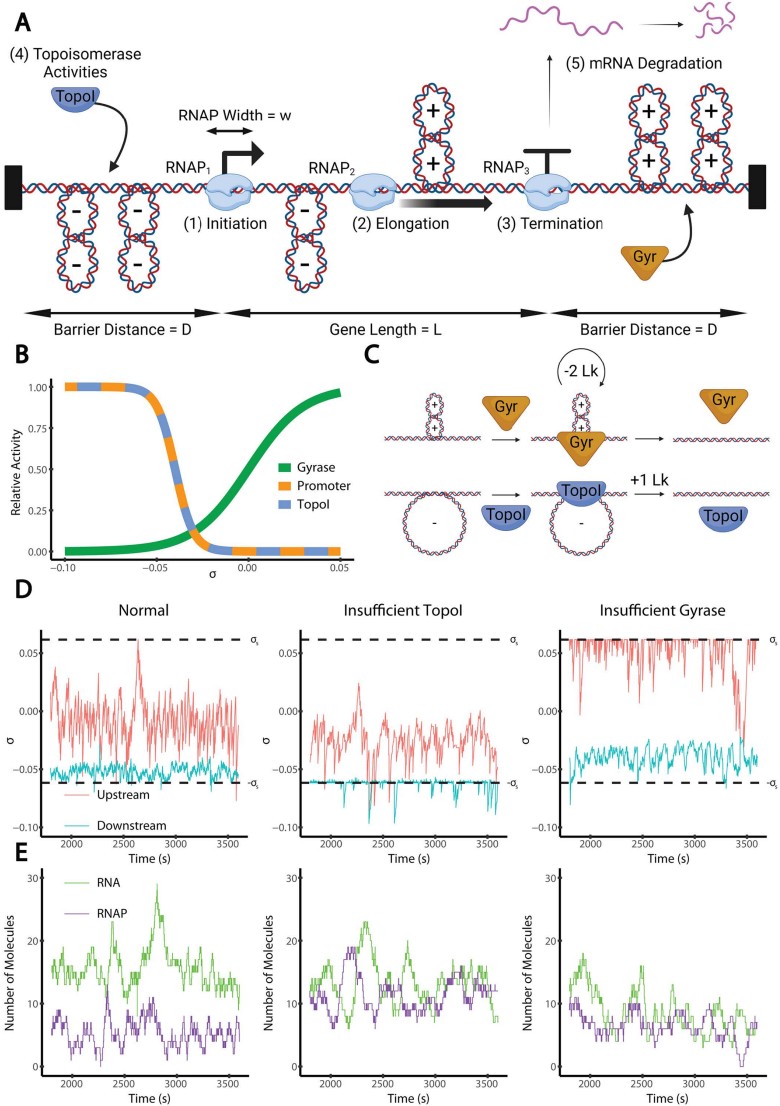

**Fig 1. Model of supercoiling-coupled transcription.** (A) Model setup. For transcription, we consider the following fives aspects: (1) initiation, a single step process dependent on SC; (2) elongation, which introduces positive and negative supercoils downstream and upstream respectively; it occurs at a constant rate except when stalled under extreme supercoiling levels; (3) termination, which occurs instantaneously and produces a single RNA transcript; (4) topoisomerase activities, including gyrase and TopoI; and (5) RNA degradation, which occurs via a Poisson process. The gene length, $L$, is the distance between the transcription start site (TSS) and terminator; the barrier distance, $D$, is the sequence length between the TSS and upstream topological barrier and between the terminator and downstream topological barrier. (B) Modeled relative activity (normalized to the maximum possible rate) of gyrase binding (green), transcription initiation (orange), and TopoI binding (blue) at varying SC densities ($\sigma$). (C) Gyrase preferentially binds positively supercoiled DNA and decreases the linking number by 2 per catalytic cycle. Gyrase acts processively, performing four catalytic cycles per binding event on average. TopoI preferentially binds negatively supercoiled DNA and increases the linking number by 1 per binding event. (D) Example simulation time courses of supercoiling density ($\sigma$) upstream (blue) and downstream (red) of the gene under three different conditions: normal, insufficient Topo I activity, and insufficient gyrase activity. (E) Corresponding time courses of produced RNA copy number and number of bound RNAP molecules under the three conditions. The stalling SC thresholds, $\pm\sigma_s$, are shown as black dashed lines. Under the normal condition (left, gyrase activity 0.24 Lk/kb/s, TopoI activity 0.24 Lk/kb/s), transcription maintains continuous elongation with infrequent stalling. Under the insufficient TopoI activity condition (middle, gyrase activity 0.24 Lk/kb/s, TopoI activity 0.095 Lk/kb/s), transcription frequently stalls due to accumulated negative SC. Under the insufficient gyrase activity condition (right, gyrase activity 0.095 Lk/kb/s, TopoI activity 0.24 Lk/kb/s), transcription also stalls frequently due to accumulated positive SC. Under normal conditions, the RNA copy number consistently exceeds the RNAP copy number, indicating that RNAPs quickly transcribe the entire gene. In contrast, when either topoisomerase is insufficient, the RNA copy number decreases while the RNAP copy number increases, indicating that the RNAPs are stalled during elongation and spend more time on the gene. All examples are for strong genes ($k_i = 0.2\,s^{-1}$) with intermediately distant barriers ($D = 10\,kb$). All cartoons were created with BioRender.com.

**Table 1. Summary of deterministic and stochastic model processes.** RNAP elongation, termination, and SC diffusion are simulated deterministically. Elongation occurs at a rate of $v_{RNAP}$ when the upstream and downstream SC levels are within an acceptable range, and stalls otherwise. Termination occurs instantaneously when an RNAP reaches the terminator. SC diffusion occurs instantaneously whenever topoisomerases act or RNAPs bind, unbind, or elongate. Initiation, topoisomerase binding, and RNA decay are simulated stochastically. Initiation occurs at a rate with a negatively sloped sigmoidal dependence on the local SC density, $\sigma$. This sigmoidal curve is defined by its midpoints, $\sigma_i$, and width, $\beta_i$. Gyrase binding has a positively sloped sigmoidal SC dependence defined by similar parameters $\sigma_G$ and $\beta_G$, and TopoI binding has a negatively sloped sigmoidal SC dependence defined by $\sigma_T$ and $\beta_G$. RNA decay occurs according to a Poisson process with rate $t_{RNA}^{-1}$.

| | Process | Effect | Rate | Units |
|---|---|---|---|---|
| **Deterministic** | RNAP elongation | No rotation.<br>RNAP moves 1 bp downstream unless stalled. | $v_{RNAP}$ if $-\sigma_s < \sigma^{(up)}$ and $\sigma^{(down)} < \sigma_s$; 0 otherwise. | bp.s$^{-1}$ |
| | Termination | RNAP at terminator removed; RNA produced. | Instantaneous. | – |
| | SC Diffusion | SC Equilibrates between RNAPs or topological barriers. | Instantaneous | – |
| **Stochastic** | Initiation | RNAP created at promoter | $k'_i = \dfrac{k_i}{1+exp\left(\frac{\sigma-\sigma_i}{\beta_i}\right)}$ | s$^{-1}$ |
| | Gyrase Binding | Gyrase binds upstream or downstream of gene, decreasing Lk by 2x, where x is a geometric random variable with mean $\rho_G$. | $k'_G = \dfrac{k_G}{1+exp\left(-\frac{\sigma-\sigma_G}{\beta_G}\right)}$ | bp$^{-1}$.s$^{-1}$ |
| | TopoI Binding | TopoI binds upstream or downstream of gene and increases the linking number by 1. | $k'_T = \dfrac{k_T}{1+exp\left(\frac{\sigma-\sigma_T}{\beta_T}\right)}$ | bp$^{-1}$.s$^{-1}$ |
| | RNA Decay | RNA removed. | $t_{RNA}^{-1}$ | s$^{-1}$ |

RNAP molecule, so that the downstream end of the RNAP is at the transcription start site (TSS). This newly added RNAP is indexed as RNAP$_1$, and the indices of the existing RNAPs are increased by one (RNAP$_1$ becomes RNAP$_2$, RNAP$_2$ becomes RNAP$_3$, etc.). Linking numbers are calculated and updated for the newly bound RNAP molecule and, if present, the adjacent downstream RNAP molecule. Initiation cannot occur if the promoter is occluded by an already-bound RNAP molecule, i.e., when RNAP(s) are bound and $x_1 < D + \frac{w}{2}$.

When the promoter is free, the initiation rate is dependent on the supercoiling density, $\sigma$, (Fig 1B) according to the following sigmoidal function, where $k'_i$ is the initiation rate; $k_i$ is the maximal initiation rate (under infinitely negative SC); $\sigma_i$ is the SC density at which $k'_i$ is half of $k_i$; and $\beta_i$ is the crossover width, which characterizes the sensitivity of $k'_i$ to changes in $\sigma$:

$$k'_i = \frac{k_i}{1+exp\left(\frac{\sigma-\sigma_i}{\beta_i}\right)}$$

Based on *in vitro* data on transcription, we note that initiation reaches a maximum around $\sigma = -0.05$ and declines to a very low level around $\sigma = -0.03$ to $\sigma = -0.01$ [38]. To match this curve, we therefore use $\sigma_i = -0.04$ and $\beta_i = 0.005$ except where specified otherwise.

## Elongation

At each timestep, the position $x_i$ of each RNAP molecule (*RNAP$_i$*) is increased by one bp, commensurate with a basal elongation speed of 25 bp/s [3,42], in the absence of stalling. Stalling occurs for *RNAP$_i$* at position $x_i$ if $\sigma_i^{(up)} < -\sigma_s$ or $\sigma_i^{(down)} > \sigma_s$, where $\sigma_s = 0.062$ is the stalling threshold calculated from experimental observations [3,21]. We assume that the polymerase is sufficiently drag-limited that only the DNA, and not the RNAP, rotates during elongation, meaning we hold Lk$_i^{(up)}$ and Lk$_i^{(down)}$ constant during elongation. After each translocation, supercoiling densities for the translocated and neighboring RNAP molecules are then recalculated based on their updated distances. The average elongation speed of each

RNAP molecule is calculated by dividing the gene length by the time the RNAP molecule spends from initiation to termination. Thus, a frequently stalled RNAP molecule will have a slow elongation speed.

We note that several approaches have been taken to model RNAP stalling. While our work, as in [21], considers the upstream and downstream torques to act independently, others base stalling on the net torque experienced by RNAP, defined by a function of the difference between upstream and downstream superhelical densities [25,30]. Still others use a combination of both approaches [28]. Considering the upstream and downstream torques independently and deterministically has the advantage that the upstream and downstream torques required to stall RNAP have been individually determined by experiment [3,28], but no similar data exists for torques exerted simultaneously upstream and downstream. Still, future work should isolate the consequences of these varied models of RNAP elongation, including varied deterministic and stochastic stalling criteria.

## Termination and RNA

Termination occurs deterministically and instantaneously when an RNAP molecule reaches the termination site, i.e., $x_k = L + D$. Each termination event increases the RNA copy number by one. Upon termination, $RNAP_k$ is removed. If there are additional RNAP molecules bound to the gene, the linking number of the new downstream RNAP molecule after removal is updated to reflect the merged domains.

We assume that RNA molecules are degraded by a Poisson process, where each RNA molecule is assigned a lifetime drawn from an exponential distribution with expected lifetime $t_{RNA}$, after which it is removed from the simulation. We use $t_{RNA} = 120\,s$, on par with past experimental measurements [42].

We quantify transcriptional noise by calculating the Fano factor, calculated from the distribution of RNA copy numbers at each time point of the simulation by dividing the variance by the mean. This statistic is the natural choice for discriminating between Poisson and bursty transcription because a Poisson process has a Fano factor of one regardless of the mean level. Transcriptional bursting (bunched RNA production during "on" periods separated by inactive "off" periods) is characterized by a Fano factor greater than one.

## Topoisomerases

Topoisomerase activities are simulated stochastically. Gyrase and TopoI bind nonspecifically upstream and downstream of the gene. TopoI binds to the DNA at a rate determined by a decreasing sigmoidal function of SC defined by parameters $k_T$, $\sigma_T$, and $\beta_T$, where $k_T$ is the TopoI binding rate under optimal conditions, $\sigma_T$ is the SC density where gyrase binds at half its maximum rate, and $\beta_T$ is the crossover width of the SC dependency. The dependence of TopoI binding on SC density has not been fully determined experimentally. Therefore, except when noted otherwise, we take $\sigma_T = \sigma_i = -0.04$ and $\beta_T = \beta_i = 0.005$ on the grounds that both promoter opening and TopoI activity require DNA melting [10], as depicted in Fig 1B. These values approximately match existing indirect estimates [43]. However, a dependence on the DNA sequence is also expected and might be characterized in future studies (with possibly significant impact on the interplay of topoisomerase I with transcription initiation). We did not model any direct interaction between TopoI and RNAP. Gyrase binds at a rate determined by an increasing sigmoid function of SC with analogous parameters $k_G$, $\sigma_G$, and $\beta_G$. In line with available experimental data [44,45], we take $\sigma_G = 0$ and $\beta_G = 0.015$, also shown in Fig 1B.

When a TopoI molecule binds to a DNA stretch, it increases the linking number by one and unbinds instantaneously. This linking number is tracked by both adjacent RNAP molecules and the domain linking number ($Lk_{dom}$).

When a gyrase molecule binds to a DNA stretch with a threshold of $\sigma > \sigma_\rho = -0.11$, it can perform multiple catalytic cycles before unbinding, as previously shown experimentally [46]. Gyrase processivity is simulated by drawing the number of catalytic cycles from a geometric distribution with mean $\rho_G = 4$ (obtained by fitting the distribution from [46]), where $\rho_G$ describes the average number of catalytic cycles per binding event. We then cap the number of cycles such that gyrase

never acts on DNA with $\sigma < \sigma_\rho = -0.11$; this value has been shown experimentally to be the lower bound on the SC level that gyrase can bind [44]. This procedure is equivalent to simulating successive catalytic cycles following the experimentally observed distribution of linking number changes introduced by gyrase [44,46]. Each catalytic cycle decreases the linking number by two, which is handled the same way as with TopoI. When reporting "topoisomerase activities" for plotting purposes, we report $2k_G\rho_G$ as the gyrase activity and $k_T$ as the TopoI activity to ensure comparable rates in units of Lk/kb/s.

## Simulation

Our model is implemented in Python 3.9.18 using NumPy 2.1.3. Analysis and plotting were performed in R 4.3.1. Initiation, topoisomerase activities, and RNA decay are simulated stochastically (Table 1) by randomly selecting the waiting time before each upcoming event from the appropriate time distribution. Within each 0.04 s timestep (the time for an RNAP molecule to elongate by one bp), we resolve any stochastic events scheduled to occur before simulating elongation and termination deterministically. Supercoiling levels are updated instantaneously whenever any SC-altering event occurs. Sections of simulation code were taken from [21], and new sections were introduced to account for our treatment of initiation, topoisomerase activities, and RNA degradation. Parameter choices are indicated in Table 2. All code for our simulations and analysis is available on GitHub (DOI: https://doi.org/10.5281/zenodo.14586118).

## Comparison to previous models

The main additions over existing works are the description of (nonspecific) topoisomerase activity, for which we employ a gradual response function to local SC, and the introduction of gyrase processivity, based on experimental evidence [44,46,47]. Compared to [21], we do not add a sequence-specific term in topoisomerase activity. Compared to [16,28], we do not consider the possibility for RNAP to rotate during elongation. While several of these modeling ingredients remain subject to improvements, we verified that the main conclusions of this study are robust to minor changes in choices of functions and parameters regarding initiation and topoisomerase activities. For example, we tested both logistic and step

**Table 2. Model parameter definitions and values.**

| Symbol | Parameter | Value(s) Used |
|---|---|---|
| $L$ | Gene length | [500 bp, 1 kb] |
| $D$ | Barrier distance | [500 bp, 100 kb] |
| $\sigma_{start}$ | Initial Supercoiling Density | -0.055 |
| $k_G$ | Gyrase Binding Rate | [0, 6 Mb$^{-1}$.s$^{-1}$] |
| $\sigma_G$ | Gyrase SC Dependence: Midpoint [44,45] | 0 |
| $\beta_G$ | Gyrase SC Dependence: Crossover Width [44,45] | 0.015 |
| $\rho_G$ | Average Gyrase Cycles Per Binding Event [46] | 4 |
| $\sigma_\rho$ | Gyrase SC Dependence: Lower Threshold [44] | -0.11 |
| $k_T$ | TopoI Binding Rate | [0, 47 Mb$^{-1}$.s$^{-1}$] |
| $\sigma_T$ | TopoI SC Dependence: Midpoint | -0.04 |
| $\beta_T$ | TopoI SC Dependence: Crossover Width | 0.005 |
| $k_i$ | Initiation Rate | [0, 0.2] s$^{-1}$ |
| $\sigma_i$ | Initiation SC Dependence: Midpoint [17] | [-0.04, -0.06] |
| $\beta_i$ | Initiation SC Dependence: Crossover Width [17] | 0.005 |
| $v$ | RNAP Velocity [3,42] | 25 bp.s$^{-1}$ |
| $\sigma_s$ | Stalling Threshold [3,21] | 0.062 |
| $t_{RNA}$ | RNA lifetime [42] | 120 s |

functions for initiation and topoisomerase binding rates and did not observe qualitatively different conclusions. We chose to adopt the logistic functions because the width of the curve did not modify the overall results (within a reasonable range of values). Therefore, the use of logistic functions enhances the stability of our simulations in response to small changes in SC levels without significantly increasing the model's parameterization.

## Transcription of a freely rotating plasmid without topoisomerases

We simulated the case of a gene transcribed on a circular plasmid without topoisomerases (as in a usual *in vitro* setup), where positive and negative SC introduced by elongation diffuse around the plasmid and annihilate one another. This annihilation leads to a constant global SC level. We simulated short genes ($L = 500$ bp) with a strong promoter ($k_i = 0.4$ s$^{-1}$) that opens at different SC densities ($\sigma_i = $ -0.06, -0.05, and -0.04). Since our simulation scheme was designed for linear DNAs, we approximated a circular plasmid by using extremely distant barriers ($D = 10^9$ bp), resulting in negligible SC accumulations, which mimic the annihilation of positive and negative supercoiling around a plasmid. While this approximation differs from the desired "reflective boundaries" simulation method, it is computationally cost-effective, and we confirmed that the SC level fluctuations caused by RNAP elongation were less than 0.0003, insignificant in the context of our model to cause RNAP stalling.

For each condition, we ran 21 simulations, varying the SC density from $\sigma_{start} = $ -0.1 to $\sigma_{start} = 0$ in steps of 0.005. Simulations were run for $10^6$ timesteps (11.11 hours simulated time), using approximately 1.7 CPU minutes each. To calculate the transcription rate (number of RNA/time), we divided the number of termination events (i.e., the number of produced RNA) by the total time of each simulation for individual plasmid topoisomers. To compare the simulations with transcription levels from a distribution of topoisomers, we assumed a normal distribution of linking numbers with a standard deviation of 0.004, a typical value for plasmids extracted from cells and measured by agarose-chloroquine gels [17]. We then calculated the transcription rates for distributions of plasmid topoisomers with mean SC densities ranging from -0.1 to 0.

## Transcriptional regulation of a topologically isolated gene

We simulated transcription of a gene on linear DNA flanked by topological barriers in the presence of topoisomerases, mimicking a topological domain *in vivo*. In all simulations, we used a gene length of $L = 1$ kb. We varied four variables between these simulations: promoter strength ($k_i$), barrier distance ($D$), TopoI binding rate ($k_T$), and gyrase binding rate ($k_G$). Promoter strengths were chosen based on the expected number of RNAP molecules on the gene, assuming conditions permissive of initiation and elongation to simulate different transcription regimes: a weak promoter ($k_i = 0.008$ s$^{-1}$, average RNAP molecules per gene $= 0.32$) simulates intermittent transcription, a moderate promoter ($k_i = 0.05$ s$^{-1}$, avg. RNAP molecules per gene $= 2$) simulates mostly continuous transcription with occasional breaks, and a strong promoter ($k_i = 0.2$ s$^{-1}$, avg. RNAP molecules per gene $= 8$) simulates continuous transcription. As a control, we also consider the case of no transcription. We consider barrier distances of $D = 1$ kb (close barriers), 10 kb (intermediate barriers), and 100 kb (far barriers).

For each barrier distance, we calculate the steady-state binding rates of both TopoI and gyrase such that TopoI removes the upstream negative SC and gyrase removes the downstream positive SC at the same rate that SC is introduced by elongation, assuming TopoI and gyrase bind at their maximum rates. We term these values $k_T^*$ and $k_G^*$, and we use them to guide our parameter choices, as we would expect different behaviors when $k_T < k_T^*$ and $k_T > k_T^*$, and likewise when $k_G < k_G^*$ and $k_G > k_G^*$. We simulate each combination of $k_T$ and $k_G$ achieved by varying $k_T$ from 0 to $2k_T^*$ in increments of $0.1k_T^*$ and $k_G$ from 0 to $2k_G^*$ in increments of $0.1k_G^*$. Note that $k_T^*$ and $k_G^*$ are inversely proportional to the barrier distance $D$ (since we assume non-specific binding of topoisomerases). We run additional sets of simulations with 10-fold increased and decreased topoisomerase activities compared to those described above to enable comparisons between the effects of altering barrier distance and topoisomerase activities. To simulate the effects of topoisomerase inhibition, we run simulations with a 5-fold decreased gyrase or TopoI activity separately in the conditions of 10 kb and 100 kb barrier distances.

Finally, we run simulations to match the experimental setup of [21]: we consider an upstream barrier distance of 250 bp and a downstream barrier distance of 320 bp; we then incrementally increase the upstream barrier distance to 3200 bp or the downstream barrier distance to 3408 bp independently, calculating the ratio in expression strength (Far/Close).

We therefore generated 6237 total simulations (11 TopoI conditions, 11 gyrase conditions, 4 promoter conditions, 3 barrier conditions, and 3 topoisomerase conditions for the primarily analysis; 11 TopoI conditions, 11 gyrase conditions, 3 promoter conditions, and 2 barrier conditions for the individual topoisomerase inhibition simulations; 11 TopoI conditions, 11 gyrase conditions, 3 promoter conditions, and 3 barrier conditions for varying the upstream/downstream distances independently; and 3 promoter conditions and 22 barrier conditions for the fine-grained varying of upstream/downstream distances). For each simulation, we run $2 \times 10^6$ timesteps (22.22 hours simulated time), utilizing approximately 4 CPU minutes per simulation. In most cases, we disregard the first 45,000 timesteps (30 minutes) to focus on the steady-state behavior of transcription (in cases with 100 kb barrier distances, we had to remove 60 minutes of data, as the simulations took longer to reach the steady state). To isolate the impact of topological variables on transcription from the strength of the promoter, we calculate a normalized transcription rate by dividing the observed average transcription rate by the maximal promoter initiation rate ($k_i$). We also calculate a "free promoter fraction", representative of the proportion of time steps during which the promoter is not blocked from binding by an RNAP. Low free promoter fraction indicates that transcription is repressed by slow elongation, as RNAP does not clear the promoter in a timely manner. This value is calculated by dividing the observed transcription rate by the mean value of the effective initiation rate given the promoter SC density ($k_i'$). In our model, all RNAPs that initiate eventually produce an RNA transcript, and initiation is determined exclusively by the promoter SC density and the occupancy of the promoter by RNAP. Hence, this value isolates the effect of promoter occupancy and characterizes the fraction of time during which the promoter is free for RNAP binding.

## Results

We have developed a model of supercoiling-coupled transcription of a single gene within a topological domain. Our detailed modeling choices and discussed in the Methods section and visualized in Fig 1. Briefly, the model incorporates both stochastic and deterministic components. At each discrete timestep ($\Delta t = 0.04$ s), the promoter can experience transcription initiation, modeled as a one-step, stochastic, SC-dependent process; all bound RNAPs undergo deterministic elongation that can be stalled by extreme SC levels; and termination occurs instantaneously upon reaching the terminator. The SC level is affected by elongation (assuming no rotation of RNAP), diffuses instantaneously between topological barriers (RNAPs or domain boundaries), and is modulated by stochastic gyrase and topoisomerase I activities promoted by more positive or negative SC levels, respectively. RNA decay is modeled as a stochastic Poisson process. All steps of the model are listed in Table 1.

### Simulation of in vitro transcription recapitulates optimal expression at physiological negative superhelical levels

To validate our model, we first compared its predictions to *in vitro* transcription data obtained using circular plasmids of different superhelical densities incubated with RNAP. These data have been a major source for identifying the SC-sensitivity of bacterial promoters [36–38]. Because supercoils generated by transcription can diffuse and annihilate each other in a circular freely rotating plasmid, the SC level can be approximated as constant in such templates. Previous experiments have shown that a gene typically transcribes maximally at an optimal negative SC level and exhibits decreased transcription when the negative SC level is too low or too high (S1 Fig). However, no stochastic model has reproduced this non-monotonic behavior, a basic yet crucial observation of transcription in response to negative SC.

In our simulations, transcription ceased completely at very negative SC densities ($\sigma < \sigma_s$) due to RNAP stalling (Fig 2A, left side of the red vertical line), and decreased continuously when the SC density increases above this threshold (Fig 2A, right side of the red vertical line), due to a reduced initiation rate. This transcriptional response to the SC density of the

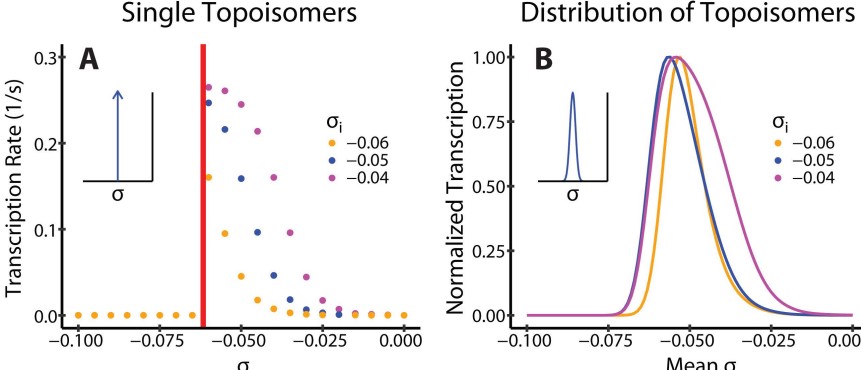

**Fig 2. Transcription on Freely Rotating Plasmids.** We simulate transcription on circular plasmids in the absence of topoisomerases, mimicking *in vitro* transcription assays. (A) Transcription rates of three promoters that open at different $\sigma_i$, simulated on plasmids with different SC densities, $\sigma$, but with only a single topoisomer. The RNAP stalling threshold $\sigma_s$ is indicated in red. (B) Transcription rates of the same three promoters were simulated on plasmids with normally distributed SC densities with multiple topoisomers, which mimics typical topoisomer distributions of plasmids extracted from cells [36–38]. Inset plots show the distribution of SC densities for plasmids of a given mean SC level (a Dirac delta function for A, and a Gaussian distribution for B).

plasmid holds true for promoters with different initiation dependencies on SC ($\sigma_i$, Methods) (Fig 2A, compare data points of different colors).

To further compare with experiments performed on plasmid samples of different mean SC densities and with a distribution of topoisomers (with different linking numbers, as commonly extracted from cells), we computed the total expression level from the weighted averages of different templates (each corresponding to one plasmid topoisomer) and calculated the effective transcription rate of samples with different mean SC densities. Fig 2B shows the resulting continuous SC-dependency where transcription is maximized by a moderately negative SC density, and decreased in either direction from this optimum, recapitulating previous experimental results [36–38]. This result validates our model for different promoters, suggesting that it can faithfully recapitulate the non-monotonic dependence of transcription on negative SC levels. The very sharp decrease towards very negative SC levels suggests that in *in vitro* transcription systems, RNAP might experience weaker frictional drag due to the absence of translation and a less crowded environment than that simulated in our model.

### Regulatory effects of topoisomerases' activities depend on promoter strength

Next, we simulated transcription on linear DNAs with topological barriers in the presence of topoisomerase activities. This situation mimics typical transcription occurring in bacterial chromosomal topological domains. We simulated topological barriers at distances of 10 kb flanking a gene of a typical size, 1 kb, leading to a 21 kb topological domain, in line with experimentally reported sizes of chromosomal domains in *E. coli* [48]. TopoI and gyrase molecules can bind upstream or downstream of elongating RNAPs, depending on their binding rates and the local SC densities (see Methods). In the absence of transcription, the relative activities of the two topoisomerases maintain a domain-wide (homeostatic) SC level of about $\sigma$ = -0.06 to -0.02, consistent with that typically measured in bacterial cells (S2A Fig). Meanwhile, transcription substantially alters the spatiotemporal distribution of SC in the domain (S2C Fig) while maintaining a similar average domain-wide SC density (S2B Fig).

In our simulations, we define the transcription rate as the number of RNA molecules generated per unit time, normalized to the maximal transcription rate that the promoter can achieve (Fig 3, column i). Note that, as we do not consider premature termination in our simulations, all RNAP molecules that initiate will eventually produce a transcript. Therefore, the transcription rate should be equivalent to the transcription initiation rate and independent of the elongation speed

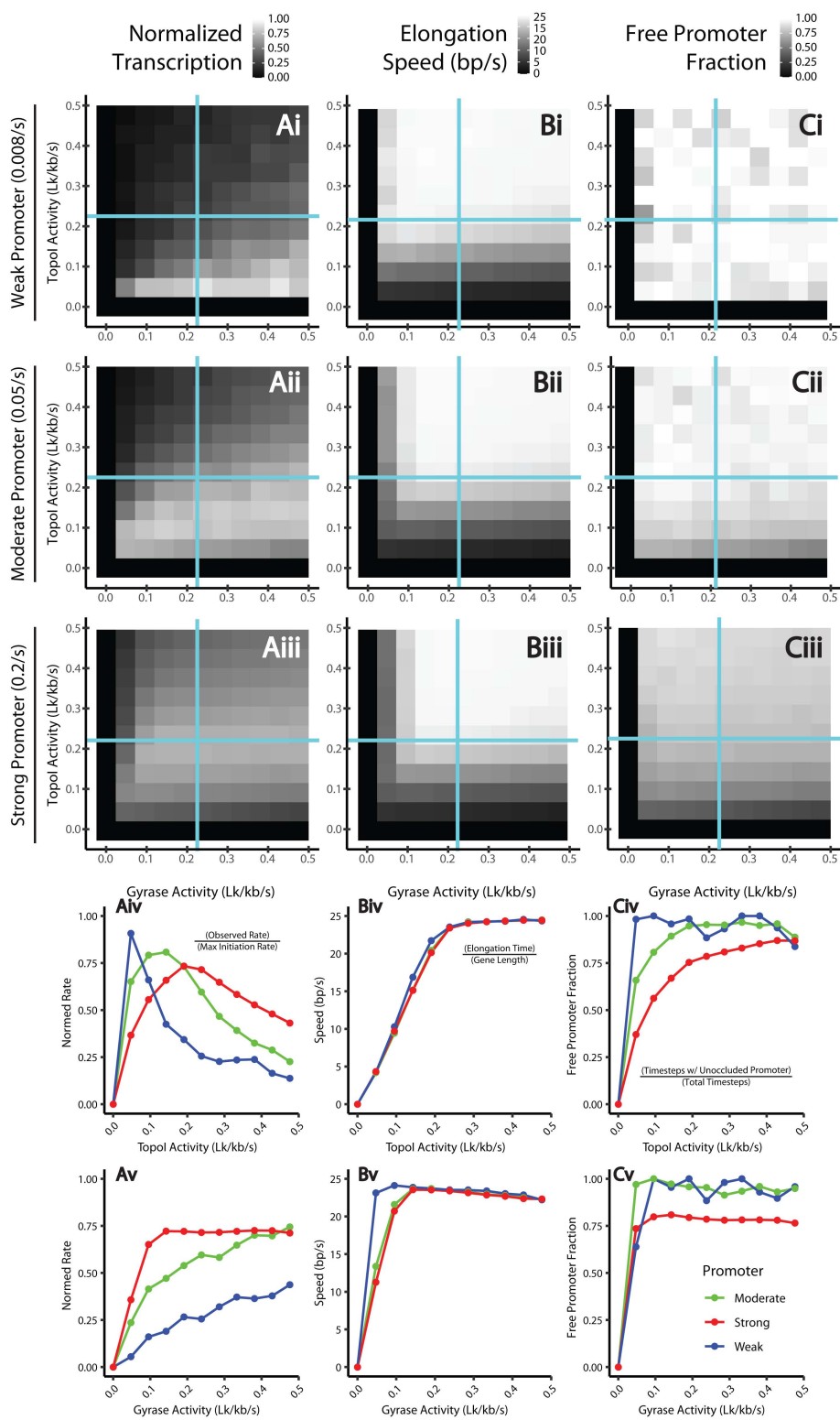

**Fig 3. Regulatory effects of topoisomerases on transcription. (column A)** Mean normalized transcription rate, $k_{obs}/k_i$; i.e., the expression strength normalized by the promoter strength. **(column B)** Mean elongation speed. **(column C)** Free promoter fraction, i.e., the proportion of timesteps during which the promoter is not sterically occluded by a bound RNAP molecule. We consider a 1 kb gene flanked by 10 kb distances to topological barriers. We

vary topoisomerase activities ($k_G$ from 0 to $2k_G{}^*$ and $k_T$ from 0 to $2k_T{}^*$, where the $k_G{}^*$ and $k_T{}^*$ and the topoisomerase activities calculated to remove the supercoiling introduced by continuous elongation) and plot heatmaps of each of these variables calculated for weak (**row i**, $k_i = 0.008\,\text{s}^{-1}$), moderate (**row ii**, $k_i = 0.05\,\text{s}^{-1}$), and strong (**row iii**, $k_i = 0.02\,\text{s}^{-1}$) promoters. We also fix either $k_T = k_T{}^*$ (**row iv**) or $k_T = k_T{}^*$ (**row v**) and vary the other topoisomerase's activity (i.e., we trace along the blue lines in rows i-iii) and plot each statistic described above.

(Fig 3, column ii). However, an RNAP molecule may stay at the promoter and occlude the binding of the next RNAP molecule to the promoter if it is stalled at the promoter due to an unfavorable SC level or blocked from moving past the promoter due to piled, stalled RNAP molecules in the gene body. Therefore, a slow elongation speed, which reflects stalling during initiation and/or elongation, could impact the transcription rate through the availability of the promoter to the next RNAP molecule. To illustrate this point, we calculated the free promoter fraction, which is the proportion of time steps during which the promoter is free for RNAP binding (Fig 3, column iii). As such, the free promoter fraction is a direct indicator of the impact of RNAP's stalling on the transcription rate, with lower free promoter fractions indicating transcription repression caused by stalled RNAP. Fig 3A-C shows the values of these statistics for all topoisomerase conditions we tested, while rows Fig 3D-E isolate the effects of either TopoI or gyrase activity.

From these simulations, we observe that TopoI and gyrase impact promoters of varied strengths in a qualitatively different manner, with each promoter reaching its maximal transcription rate at a different TopoI activities (Fig 3D) and having different sensitivities to gyrase activities (Fig 3E). Specifically, for a weak promoter, high gyrase activity (leading to more negative SC densities) increases the transcription rate (Fig 3A and 3Ei), while a high TopoI activity (leading to less negative SC density) inhibits the transcription rate (Fig 3Ai and 3Di). Interestingly, the highest transcription rate of the weak promoter does not correspond to the fastest elongation rate (compare Fig 3Ai to 3Bi). In other words, slow elongation speed, which signals frequent RNAP stalling, does not appear to reduce the overall transcription rate for the weak promoter. However, as the free promoter fractions across different topoisomerase conditions remain high (Fig 3Ci), it indicates that the weak promoter is limited by transcription initiation but not RNAP stalling. The promoter is maximally transcribed under topoisomerase conditions that result in the most negative SC (low TopoI and high gyrase activities), which facilitate promoter opening and elongation to increase the transcription rate. Under this condition, RNAP stalling due to insufficient topoisomerases' activities still occurs; however, it does not substantially affect the transcription rate because the initiation rate is limiting.

As we increase the promoter strength, we observe markedly different behaviors. Moderate and strong promoters reach their maximum transcription rate at moderate TopoI activities and moderate to high gyrase activities. Increasing TopoI activities increases the elongation speed until minimal stalling occurs, at which point the free promoter fraction is near one. Past this point, the addition of TopoI increases the SC density at the promoter, thus slowing initiation and, therefore, transcription. As gyrase activity increases, a maximum free promoter fraction is reached more quickly. However, additional gyrase continues to increase transcription by decreasing SC density at the promoter, facilitating initiation. The strong gene requires more TopoI activity to reach a free promoter fraction of one, suggesting that strong genes are more sensitive to RNAP-stalling-mediated transcription repression than weak to moderate genes.

Taken together, our simulations show that the transcription rate can be regulated by combined Topo I and Gyrase activities through the availability of the promoter. A high free promoter fraction, such as that in the case of the weak promoter, signals that the promoter SC level is limiting for RNAP to initiate transcription. Meanwhile, a low free promoter fraction, such as that in the case of the strong promoter, suggests that RNAP stalling is limiting by making the promoter unavailable for the next RNAP molecule. Specifically, in conditions where the SC-dependent initiation rate, $k_i{}'$, is low, the free promoter fraction is high, and transcription is regulated primarily by the promoter SC level for initiation. Therefore, transcription is enhanced by increasing gyrase activity and decreasing TopoI activity to increase negative SC for promoter opening. As $k_i{}'$ increases, the increased frequency of RNAP binding means that the free promoter fraction becomes sensitive to RNAP stalling. Thus, the transcription rate becomes highly dependent on the presence of adequate levels of both TopoI and

gyrase to keep up with the SC introduced by elongation to avoid stalling. With high $k_i'$, the favorability of the promoter SC level for initiation becomes relevant only when comparing multiple conditions in which TopoI and gyrase are sufficient for avoiding stalling. In general, fast initiation leads to transcription being dependent on RNAP stalling, while slow initiation leads to transcription being dependent on the promoter SC level.

### Effects of topological domain size are mediated by the binding of topoisomerases

We next investigated the dependency of transcription on topological domain size. Specifically, we reduced and length-ened barrier distances to investigate the effects of introducing and resolving topological constraints. We keep the same gene length (1 kb) and promoter strengths and show the log2-fold change in transcription rate in response to reducing the barrier distances from 10 kb to 1 kb as a function of TopoI and gyrase activities (Fig 4A). This results in changes to tran-scription that are more positive for weaker genes and high TopoI conditions. We note that these effects are similar to the effects of reducing the topoisomerase activities 10-fold (Fig 4B) and are ameliorated by simultaneously decreasing the barrier distance and increasing the topoisomerase activities 10-fold (Fig 4C). This relationship between changes to barrier distances and topoisomerase activities was expected as in our model, topoisomerases act nonspecifically, meaning that proportionally fewer binding events of topoisomerases occur on the smaller domain. This shortening of barrier distances mimics cells in which chromosomal domains form in the presence of the same cellular pool of topoisomerases.

When we increase the barrier distance from 10 kb to 100 kb, we note that at moderate to high TopoI levels, weaker genes are enhanced while stronger genes are repressed; this trend reverses at low TopoI activities (Fig 4D). This change mimics the opening of a topological domain. This effect is replicated by increasing topoisomerase activities 10-fold while holding the barrier distance constant (Fig 4E) and ameliorated by simultaneously increasing barrier distances and decreasing topoisomerase activities 10-fold (Fig 4F). The full data for this analysis, including all simulated topoisomerase conditions, are shown in S4 Fig [49–51].

Taken together, these observations suggest that the size of a topological domain imposes an additional layer of tran-scription regulation, primarily through the availability of topoisomerases in the domain. The formation of a smaller topolog-ical domain reduces the number of topoisomerase binding events, altering the local SC level relevant to the genes within the domain. Strong genes in smaller domains are hence repressed by the inability of topoisomerases to keep up with the SC introduced by elongation. Meanwhile, weak to moderate genes respond differentially depending on the combination of the two topoisomerase activities. On large domains, transcription is favored by low TopoI and high gyrase levels, as the effective topoisomerase activities increase, resulting in a high free promoter fraction. The interaction between promoter strength, topoisomerase binding rates, and topological context must therefore be considered to quantitatively understand a gene's regulation, consistent with [21].

### SC modulation of the off-state leads to transcriptional bursting

Transcriptional bursting is the phenomenon of a gene possessing multiple states in which transcription occurs at differ-ent rates (typically on or off) [52]. The bursting phenomenon results in transcriptional noise greater than that of a ran-dom birth-and-death Poisson process, typically inducing variable gene expression in an isogenic cell population, which enhances its resilience to environmental uncertainty [34]. Since SC modulates transcriptional initiation in our model, we wished to understand whether it could serve as a generator of transcriptional bursting in the absence of additional factors such as stochastic binding of regulators or other proteins affecting the promoter accessibility. We therefore analyzed the simulations to characterize the conditions that result in transcriptional bursting and identify underlying mechanisms.

The level of noise can be quantified using the normalized variance of expression (variance divided by the square of the mean). We computed this normalized variance at all promoter strengths, topoisomerase binding rates, and barrier distances, and observed that this parameter decreases with the expression level, making it challenging to compare simulations involving different promoter strengths (Fig 5A). Rather, we computed Fano factors (variance divided by the

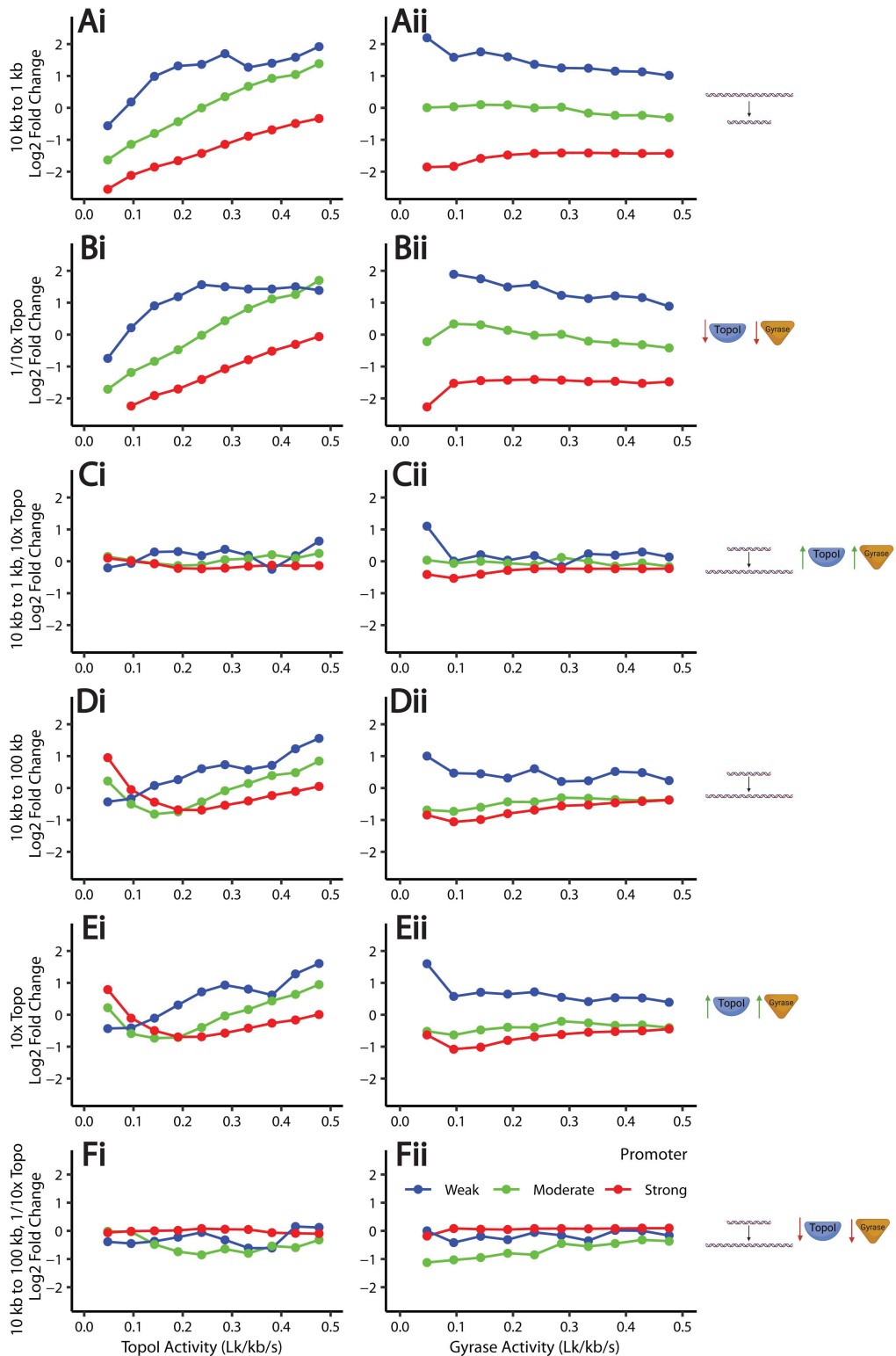

**Fig 4. Effects of barrier distances mediated by topoisomerase binding.** We fix either gyrase **(left)** or TopoI **(right)** activity at the steady-state value (blue lines in Fig 3, computed as the value sufficient to remove the SC induced by elongation), and vary the other topoisomerase's activity. We plot the log2 fold change in the expression rate in response to (A) reducing the barrier distances from 10 kb to 1 kb, (B) reducing nonspecific topoisomerase

activities 10-fold, (C) reducing the barrier distances from 10 kb to 1 kb and increasing topoisomerase nonspecific activities 10-fold to compensate, (D) increasing the barrier distances from 10 kb to 100 kb, (E) increasing the nonspecific topoisomerase activities 10-fold, and (F) increasing the barrier distances from 10 kb to 100 kb and decreasing the nonspecific topoisomerase activities 10-fold to compensate. Cartoons were created with BioRender.com.

mean) for the same simulations (Fig 5B). We note that the Fano factor is not scale-free (i.e., the Fano factor depends on the transcription strength), however regardless of scale a Fano factor of one indicates Poissonian expression (i.e., constant initiation rate), whereas values greater than one indicate transcriptional bursting. We observed Fano factor values ranging from 0.7 to 4.8, with bursty transcription appearing in diverse promoter and topological conditions. The datapoints separate horizontally along the axis of mean RNA. Copy numbers belong to three clouds corresponding to simulations with weak, moderate, and strong promoters, respectively. For most conditions, the distance to barriers has little impact on transcription burstiness. However, promoter strength appears to play a role. Weak promoters' transcription is close to Poissonian (Fano factors close to 1, squares, Fig 5B), with moderate promoters exhibiting consistently higher Fano factors (about 1–2, triangles, Fig 5B), indicative of moderate burstiness. We observed a large number of simulations exhibited Fano factors below 1, i.e., sub-Poissonian, suggesting that transcription becomes less random because initiation events are more regularly spaced by the time RNAP molecules take to clear the promoter.

To further explore transcription bursting dynamics, we consider in-depth the case of a moderate promoter ($k_i = 0.05\,\text{s}^{-1}$) with intermediate barriers ($D = 10\,\text{kb}$). This promoter strength and domain size (green triangles in Fig 5B) might be representative of some bacterial genes and exhibit weakly bursty transcription independent of barrier distance (S5 Fig), matching experimental observations of Fano factors typically between 1 and 3 for native genes [34]. In Fig 5C, we show Fano factors for simulations of a moderate gene ($k_i = 0.05\,\text{s}^{-1}$) flanked by intermediate barriers ($D = 10\,\text{kb}$) for different combinations of TopoI and gyrase activities, and chose three scenarios for detailed analyses: scenario 1, with topoisomerase activities matched to the rate of SC introduced by elongation; scenario 2, with low gyrase activity; and scenario 3, with low TopoI activity. In scenario 1, the Fano Factor is 1.6 and the steady-state topoisomerase binding rates exactly cancel the SC introduced by elongation; in scenario 2, the Fano Factor is at 2.0 and the gyrase activity is lower than that in Scenario 1; in scenario 3, the Fano Factor is 0.88 and the TopoI activity is lower than that in Scenario 1. For each scenario, we consider the distribution of the RNA copy numbers (Fig 5D), the time course of the upstream and downstream SC density (Fig 5E), and the time course of the RNA copy number and number of bound RNAP molecules (Fig 5F).

Scenarios 1 and especially 2 describe a mechanism for transcriptional bursting in which the upstream negative SC induced by elongation facilitates the binding of additional RNAP molecules at the promoter, hence facilitating more frequent initiation events. In scenario 2, apparent "on-off" transitions are observed in accordance with a higher Fano factor at 2: the very low gyrase activity resulted in a relatively high SC density at the promoter that mostly, but not completely, suppresses initiation (the "off" state) when the gene is not bound by any RNAP molecules, whereas a single transcription event turns the gene into the "on" state by reducing the promoter SC level (Fig 5E). Whenever all RNAP molecules stochastically unbind, the positive SC accumulated downstream diffuses onto the promoter, suppressing initiation and returning the gene to the off state as the existing RNA fully decays. The off-state lasts on the order of 10 minutes, in line with experimental data [53]. Scenario 1 demonstrates on- and off-states by the same mechanism, but the off-state is shorter as the relatively high gyrase activity results in a steady-state SC level more favorable for initiation before previous RNA molecules are degraded in the off-state (Fano factor of 1.60). This mechanism implies a cooperative mode of multiple RNAP molecules as predicated previously and is interesting for its alignment with the experimental observation that transcriptional bursting is related to low gyrase expression [34]. Scenario 2 displays a greater Fano factor than scenario 1 (2.0 vs 1.6) because the greater steady-state SC level in the absence of bound RNAP molecules results in less frequent RNAP binding and a more prolonged off-state.

In scenario 3, we do not observe transcriptional bursting; in fact, we observe a sub-Poisson Fano factor (0.88). The SC level is strongly negative, almost always sustaining efficient initiation, hence a high transcription level, but so much so that

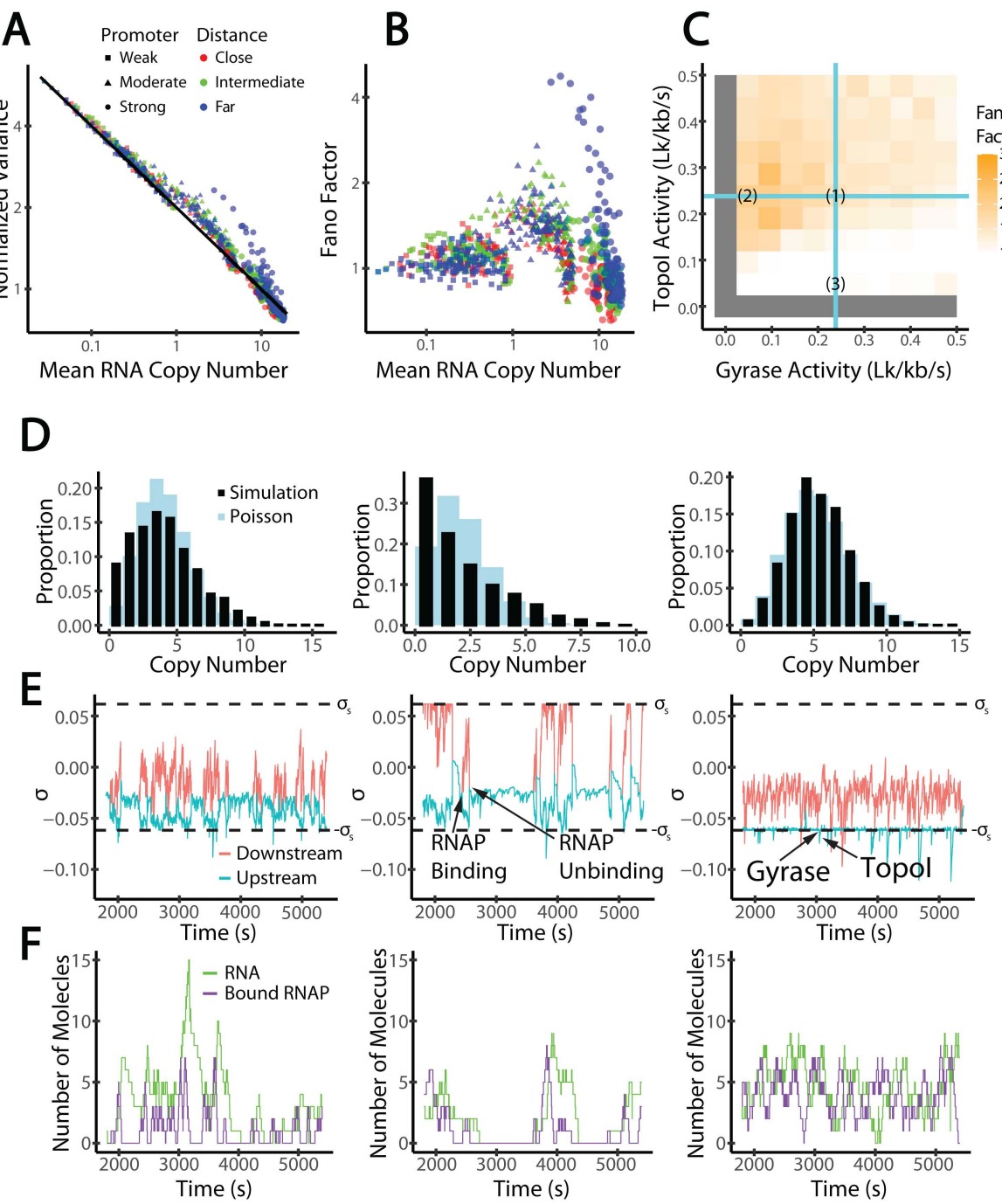

**Fig 5. Mechanism of Transcriptional Bursting.** (A) Normalized variance (noise) and mean RNA copy numbers for simulations run using varied promoter strengths (squares, $k_i = 0.008\,s^{-1}$, moderate: triangles, $k_i = 0.05\,s^{-1}$, strong: circles, $k_i = 0.2\,s^{-1}$), barrier distances (close: red, $D = 1\,kb$, intermediate: green, $D = 10\,kb$, far: blue, $D = 100\,kb$), and topoisomerase binding rates (one point per simulation). (B) Fano factors and mean RNA copy numbers for the same simulations. (A-B) are shown on a log-log scale. (C) Fano factors for a moderate gene ($k_i = 0.05\,s^{-1}$) flanked by intermediate barriers ($D = 10\,kb$). Squares correspond to individual simulations with varied topoisomerase activities. Cyan lines denote the steady-state topoisomerase activities calculated to eliminate the supercoils generated by elongation. Scenarios (1), (2), and (3) are analyzed in the remainder of the figure. (D) Distributions of RNA copy numbers (black) throughout one simulation compared with the Poisson distribution with the same mean (blue). We indicate $\pm\sigma_s$ with black dashed lines. (E) Features relevant to transcriptional bursting are indicated, including an RNAP molecule binding to the gene and an RNAP molecule unbinding after completing transcription (middle) and gyrase and Topo I binding (right). Note that RNAP unbinding leads to the merging of the upstream (red) and downstream (cyan) SC domains. (F) The time course of copy numbers of RNA (green) and bound RNAP (purple) molecules.

elongation is frequently stalled due to insufficient TopoI activity. Therefore, several RNAPs are almost always bound to the gene, and the promoter never reaches an "off" state. Because these abundant RNAPs frequently block the promoter from binding additional RNAPs, initiation events are more evenly spaced than in a Poisson distribution, explaining the low Fano factor.

We conclude that transcriptional bursting is dependent on a steady-state SC level in the "off" state, which represses, although does not eliminate, initiation, while negative SC induced by elongation turns the promoter into a transitory "on" state (*i.e.*, transcription is self-activating).

### In vivo expression data suggest moderate topoisomerase regulation of transcription, dependent on species and growth conditions

Our modeling effort so far has demonstrated that changes in topoisomerase activities could lead to qualitatively different effects on transcription. It is yet difficult to predict what parameter values are relevant within live bacterial cells, since we ignore the quantitative topoisomerase activities affecting different topological domains, as well as the latter's exact locations and boundaries. We therefore confronted our simulation results with *in vivo* expression data to identify if and what simulated conditions are compatible with the observed supercoiling regulation of transcription and to possibly infer some physiologically relevant information.

As a first source of data, we compiled previously published transcriptomic datasets [13–21,39] obtained in evolutionarily distant bacterial species (S2 Table), where cells were treated with a gyrase inhibitor (fluoroquinolones or coumarins) or TopoI inhibitor (seconeolitsine). These datasets provide the quantitative response of broad sets of genes, which differ by their topological context (nucleoid-associated proteins, neighboring genes, etc.), regulators, and surrounding sequences. Since the simulations above exhibited a strong influence of transcription strength on its sensitivity to topoisomerases, and the expression strength is experimentally available (in contrast, e.g., to the topological domain size of each gene), we looked to see if a similar relationship was detectable and robust across datasets. Because the latter are heterogeneous in the type of species, culture conditions, drug dosages, treatment time, and experimental methods used, we grouped genes into expression quartiles to improve readability. Fig 6A shows typical relations between gene expression strength and transcriptional response to gyrase inhibition, for *Mycoplasma pneumoniae* and *Salmonella enterica*. In both cases, we clearly observed that genes of higher expression were associated with transcriptional repression when gyrase is inhibited, albeit with a very different magnitude in the two species (the absolute expression ratio between the first and last quartiles is around 30-fold for *M. pneumoniae* vs around 2-fold for *S. enterica*). Note that in these analyses, the expression strengths are normalized to the total transcriptional output, so it cannot be concluded that weak genes are upregulated, but rather that their share in total mRNA numbers has increased after the treatment. We therefore show the relative ratio between weakly (1$^{st}$ quartile) and strongly (4$^{th}$ quartile) expressed genes in response to gyrase (Fig 6B) or TopoI (Fig 6D) inhibition for all datasets obtained in five species.

Two main conclusions can be drawn. First, all analyzed datasets exhibit a clear tendency towards repression of highly expressed genes when either gyrase or TopoI is inhibited, a notable robustness considering the strong heterogeneity of these datasets. Second, the magnitude of the effect is quite variable across datasets, and quite moderate for most of them (as illustrated for *S. enterica* in Fig 6A), with strong genes being repressed less than two-fold compared to weak genes. Together, these two observations suggest that topoisomerases contribute to the expression of strong bacterial genes, but only in a mild manner. Their effect also seems species- and growth condition-dependent, since *D. dadantii* is more affected by TopoI inhibition than *S. pneumoniae* (possibly due to a stronger drug dosage, both grown in minimal medium) while their sensitivity to gyrase inhibition is similar, and *E. coli* responded much stronger to novobiocin (dataset index 2) than to norfloxacin (dataset index 6) in rich medium. We obtained similar conclusions with an alternative calculation based on the correlation between expression strength (without arbitrary classification into quartiles) and expression fold-change.

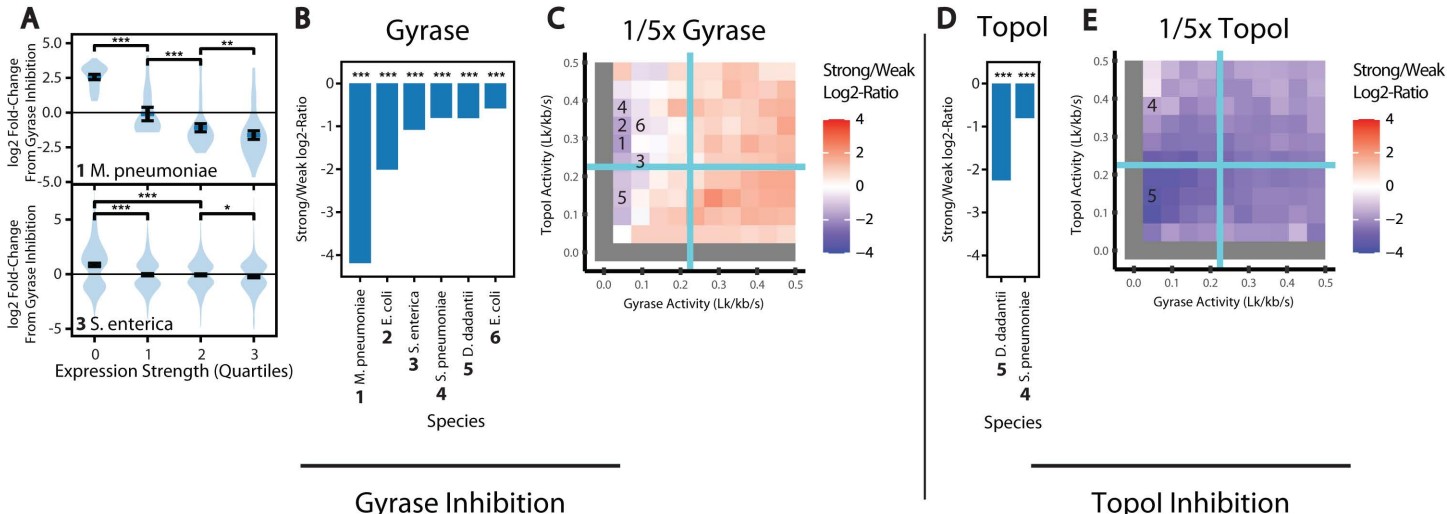

**Fig 6. Transcriptome Response to Topoisomerase Inhibition.** (A) Experimentally determined fold-change in expression of genes in *Escherichia coli* and *Mycoplasma pneumonia* in response to gyrase inhibition. Genes are split into quartiles based on expression strength. (B) Experimentally determined log2-ratio of fold-changes in strong (upper quartile) vs weak (lower quartile) expressed genes in response to TopoI inhibition. The symbol *** indicates $P < 10^{-3}$ for the associated t-test. The list of conditions is detailed in S2 Table. (C) Simulated log2-ratio of fold-changes in simulated transcription rate of strong vs weak genes ($k_i = 0.02\,\mathrm{s}^{-1}$ vs $k_i = 0.008\,\mathrm{s}^{-1}$) in response to a 5-fold reduction in gyrase activity (mimicking the action of gyrase inhibitors) for varied baseline topoisomerase activities ($k_G$ ranging from 0 to $2k_G^*$ and $k_T$ range from 0 to $2\,k_T^*$). (D) and (E) are the same as (B) and (C), respectively, but instead consider TopoI inhibition. Numbers in (C) and (E) indicate simulated topoisomerase conditions which produce log2-ratios compatible with the experimental results in (B) and (D) respectively (in several cases, we show one among several compatible conditions, see text).

We then tested if simulations could recapitulate the observed behavior, and in what parameter range. We note that while our model does not consider transcription factor binding and other complex regulatory elements in cells, it is meaningful to compare the simulation and experimental results qualitatively to identify the role and contribution of supercoiling in gene regulation. Furthermore, since the exact effect of drugs on topoisomerase activity *in vivo* is not precisely known (and likely variable in our dataset), we can only derive a qualitative comparison. We ran simulations mimicking an (arbitrary) 5-fold reduction in either enzyme, considering the log2 fold change in expression level induced in three representative genes of strong, intermediate, and weak promoter strengths (S6 Fig). Since we are interested in the relative effects of topoisomerase inhibition on genes of varied strengths, we show the ratio of these log2 fold changes between strong and weak genes in response to gyrase (Fig 6C) and TopoI (Fig 6E) inhibition (comparable to the experimental values in Fig 6B and 6D respectively) for simulations across all gyrase and topoisomerase activity values.

Contrary to naive expectations, the effect is quite variable, with gyrase inhibition resulting in a relative repression of strong genes only in a small part of the parameter space, at low gyrase activities and moderate TopoI activities (blue region of Fig 6C). In contrast, TopoI inhibition represses strong genes in almost all conditions (Fig 6E). When we consider 100 kb barriers (rather than 10 kb), we obtained essentially similar results (S6 Fig), showing some robustness to variations in simulation parameters. Although a direct comparison between our simulations and experimental values remains tentative due to the many confounding factors affecting *in vivo* transcription, we can therefore pinpoint a parameter range where the model is compatible with each dataset (see indices in Fig 6C and 6E). For two datasets, the combination of values observed with both gyrase and TopoI inhibition data (*S. Pneumoniae*, 4, and *D. dadantii*, 5) is satisfied only for specific combinations of topoisomerase activities (we indicated the most likely region for both cases with an index). The very strong repression observed in *Mycoplasma pneumoniae* (1) suggests a very low initial gyrase activity (or much stronger inhibition than in our simulation) coupled to moderate TopoI activity. For other species showing mild repression by gyrase

inhibition, the parameter range is broader (light blue region of Fig 6C) but still well-defined to a regime of weak gyrase activity (along the whole left edge of the graph).

A second source of *in vivo* data is provided by recent expression measurements of a gene in a plasmid-borne topological domain (Boulas et al. [21], whose results we reproduce in S7A Fig), showing that it is sensitive to the upstream but not the downstream distance to topological barriers (S7 Fig). Using a similar approach to that used with transcriptomics data, we tested what range of topoisomerase activity values recapitulated this asymmetric sensitivity. We found a qualitative agreement with the data under low TopoI activity, and preferentially under high gyrase activity (S7B-C Fig), i.e., a different regime than that identified from transcriptomics data (this discrepancy is addressed in the Discussion). In this regime, our model recapitulates the general observed trend, although the quantitative effect of increasing the distance does not fully recapitulate the data (S7D Fig). This difference was expected since our model includes a minimal (nonspecific) description of topoisomerase binding to DNA, which consists of a description of gyrase processivity but lacks additional terms (such as the specific binding of topoisomerase I and gyrase at certain locations) that the authors of the original study included to explain their data. We note that a comprehensive modeling of topoisomerase binding will likely involve a complex combination of DNA sequence and 3D deformations.

## Discussion

We have designed and implemented a model of supercoiling-coupled transcription, allowing a systematic exploration of regulatory effects induced by topological constraints. We used this model to directly compare simulations with diverse experimental data, from *in vitro* transcription assays to *in vivo* genome-wide responses to topoisomerase inhibition, as well as transcriptional bursting. The simulations replicated key observations from these datasets and thus provide a unifying framework towards a quantitative model of the transcription-supercoiling coupling.

To our knowledge, this is the first computational model to replicate the presence of an optimal supercoiling density observed *in vitro* with diverse promoters. While the activating effect of negative supercoiling has usually been explained by facilitated DNA opening, different hypotheses have been proposed for the inhibitory effect of strongly negative supercoiling, including the opening of alternate (less A/T-rich and therefore less favorable) DNA regions that would then thermodynamically compete with the −10 region [17]. In this work, we instead explain this drop in expression levels by a reduction in elongation efficiency. Further models may also consider possible effects of negative supercoiling on transcription initiation, in particular during promoter escape (escape rate and rate of abortive transcripts), as well as R-loop formation.

We have observed in our simulations that the effect of varying topoisomerase activity is dependent on promoter strength. Note that throughout the manuscript, we used the term "regulation" for this effect, which implies an evolutionarily selected biological control mechanism, which (to our knowledge) remains a hypothesis. While higher TopoI activities always repress weak genes due to the relaxation of promoter DNA, strong genes require an intermediate level of TopoI activity to maximize transcription, as too little TopoI activity results in RNAP stalling due to negative SC accumulation. Meanwhile, too much TopoI activity results in slowed initiation due to the removal of negative SC at the promoter. This differential regulation of weak and strong genes results from how RNAP stalling has a greater impact on the free promoter fraction for stronger genes.

Our simulations show that the introduction of topological barriers near a gene has a more inhibitory effect in strong genes, and that a reduction in the effective topoisomerase activities mediates the effects of this topological constraint. This observation is consistent with a nonspecific binding mechanism *in vivo*, as the introduction of topological constraints near a gene would result in the separation of the gene from nearby sites that topoisomerases can bind. Moreover, the observation that an inhibition of topoisomerase activities results in the preferential expression of weak genes is consistent with our analysis of transcriptomics data across the bacterial kingdom. This observation matches with previous suggestions that most bacterial genes (thousands) are transcribed at a sufficiently low level to undergo sustained transcription with little

local recruitment of topoisomerases [54], whereas the hundreds of active gyrase/TopoI molecules might concentrate in a minority of topological domains with strong transcription (especially ribosomal operons) [55].

We wished to explore if our simple model of the transcription-supercoiling coupling was able to replicate *in vivo* expression data, despite the many confounding factors that complicate the latter process, as well as our limited knowledge of the relevant parameter values. We therefore used a diverse dataset obtained from transcriptomics experiments by different groups (S2 Table), as well as a particular study involving the expression of a gene in a topologically isolated domain on a plasmid in *E. coli* [21]. By systematic parameter exploration, we were indeed able to replicate notable observations from these data, but in different parameter ranges (limiting gyrase activity for the former, limiting TopoI for the latter). This comparison remains tentative, since further improvements of the model might affect this conclusion, and more comprehensive and quantitative experimental datasets would help narrow down the relevant modeling regime (quantitative value for the gene expression strengths, precise variations in topoisomerase activities, etc.). It is therefore possible that the observed discrepancy between the two comparisons indicates the lack of an important ingredient in our modeling (see suggestions below). Within the current framework, the comparison suggests that topoisomerases affect transcription in a different manner in the plasmid employed in the latter study compared to most chromosomal genes analyzed in the transcriptomics data, with TopoI being more critical specifically in the plasmid, and gyrase in the chromosome. Possible mechanisms could include: (1) a strong impact of the employed static topological barriers (pairs of LacI proteins) inducing a regime of supercoil diffusion not encountered on the chromosome; (2) the presence of specific sequences in the plasmid, affecting topoisomerase binding; (3) other sources of differences in supercoil diffusion, e.g., due to plasmid localization or interaction with other cellular components; (4) an impact of the growth conditions (since these quite strongly impact the transcriptomics results) [21].

Our simulations also widely exhibit the experimentally observed phenomenon of transcriptional bursting, without requiring any external events of transcription factor binding/unbinding. Transcriptional bursts occur when the negative SC generated upstream of elongating RNAP molecules facilitates the binding of additional RNAP molecules, forming a DNA-embedded self-activating regulatory loop. This mechanism is dependent on the promoter obtaining a steady-state SC level that represses, but does not eliminate, initiation in the absence of already bound RNAP molecules. This Fano factor-maximizing SC level is more relaxed for strong promoters, which can exhibit extremely diverse behaviors, from sub-Poissonian to extremely bursty transcription, depending on topological conditions. Therefore, the topoisomerase binding rates that maximize the Fano factor are dependent on the promoter and topological domain length, suggesting that the behavior of a specific experimental system may not be easily controllable.

This observed behavior is largely consistent with observations in a wide range of promoters with varying expression strength and dependence on transcription factors. Fano factors in our simulations range from 0.66 to 4.75 (S5 Fig), approximately consistent with observed values [34]. A quantitative comparison will require measuring the exact number of transcripts (the mRNA number distributions measured in that study differ from those of our simulations). Chong et *al*. [34] proposed that gyrase binding introduces gyrase-bound "on" states and gyrase-unbound "off" states, and that these binding-unbinding events are responsible for transcriptional bursting, as only the gyrase-bound states can relax the downstream positive SC introduced by RNAP elongation. A previous modeling work instead proposed that this mechanism results from extrinsic looping and unlooping stochastic events, inducing different topological constraints affecting transcription [28]. However, this model had some key differences from the present model: specifically, we consider continuous dependencies of initiation and topoisomerase activities on SC density, although we do not consider RNAP rotation. Here, we propose that even in the absence of an extrinsic regulatory factor, transcription within a topological domain can be intrinsically noisy, and that, under low gyrase activity, the on/off switching is controlled by RNAP binding dynamics, i.e., the strength of the promoter, rather than gyrase binding events. This alternative explanation is supported by the observation that Fano factors are only weakly affected by gyrase inhibition [34].

Several additional aspects would be interesting to consider. As mentioned previously, it has been shown that the relative arrangements of neighboring genes play a role in how genes are regulated by topoisomerase inhibition [17] and affect

each other through supercoiling [27]. Thus, an expansion of our analysis to include multiple genes is justified. Our model is currently unable to replicate the cooperative effect of co-transcribing RNAPs with regard to elongation speed [6]. Reproducing this phenomenon would likely require refining our all-or-nothing response of elongation to SC, as already explored in other models [28–30], such as by introducing partial RNAP rotation during elongation. The ability of RNAPs to rotate varies based on the condition; for example, *in vivo* (with simultaneous translation) vs *in vitro;* RNAP molecules transcribing genes for membrane-bound proteins are more constrained due to the insertion of the simultaneously translated peptide into the membrane. Since the quantitative level of RNAP rotation remains largely unknown, it would be interesting to assess (either computationally or experimentally) how varied degrees of RNAP rotation influence the regulatory behavior of DNA topology.

## Supporting information

**S1 Table. Summary of Several Published Models of Bacterial SC-Coupled Transcription.**
(DOCX)

**S2 Table. Sources and Conditions of Expression Data.**
(DOCX)

**S1 Fig. *In vitro* SC-dependence of transcription. (A)** *In vitro* expression of several genes on plasmid samples of varied mean SC levels. The maximum expression rate for each gene is normalized to one [Auner H, Buckle M, Deufel A, Kutateladze T, Lazarus L, et al. (2003) Mechanism of transcriptional activation by FIS: Role of core promoter structure and DNA topology. J Mol Biol 331: 331–344. 10.1016/S0022-2836(03)00727–7; Borowiec JA, Gralla JD. (1987) All three elements of the *lac* p$^S$ promoter mediate its transcriptional response to DNA supercoiling. J Mol Biol 195: 89–97. 10.1016/0022–2836(87)90329–9.; Lim HM, Lewis DEA, Lee HJ, Liu M, Adhya S. (2003) Effect of varying the supercoiling of DNA on transcription and its regulation. Biochemistry 42: 10718–10725. 10.1021/bi030110t.] **(B)** An illustrative gel to show the distribution of topoisomers within each sample [Lim HM, Lewis DEA, Lee HJ, Liu M, Adhya S. (2003) Effect of varying the supercoiling of DNA on transcription and its regulation. Biochemistry 42: 10718–10725. 10.1021/bi030110t].
(EPS)

**S2 Fig. Impact of Topoisomerases and Transcription of Supercoiling. (A)** Mean SC density ($\sigma$) from simulations run in the absence of transcription at various topoisomerase binding rates. **(B)** Distribution of $\sigma$ average across the domain, upstream of the gene, and downstream of the gene for simulations run at several promoter strengths (weak: $k_i = 0.008\,\mathrm{s}^{-1}$, moderate: $k_i = 0.05\,\mathrm{s}^{-1}$, strong: $k_i = 0.2\,\mathrm{s}^{-1}$). Note that at some time intervals during which no RNAPs bind the gene, the upstream and downstream curves overlap. Topoisomerase binding rates are the steady state values $k_T^*$ and $k_G^*$, calculated to remove the SC introduced by elongation. **(C)** Time courses of the upstream (blue) and downstream (red) $\sigma$ values for several promoter strengths.
(EPS)

**S3 Fig. Regulatory effects of topoisomerases on transcription with distant barriers.** Same as Fig 3, replicated with distant topological barriers ($D = 100\,\mathrm{kb}$). **(column A)** The mean normalized transcription rate, $k_{obs}/k_i$; i.e., the expression strength normalized by the promoter strength. **(column B)** The mean elongation speed. **(column C)** The free promoter fraction, i.e., the proportion of timesteps during which the promoter is not sterically occluded by a bound RNAP molecule. We vary topoisomerase activities ($k_G$ from 0 to $2k_G^*$ and $k_T$ from 0 to $2k_T^*$, where $k_G^*$ and $k_T^*$ are the topoisomerase activities calculated to remove the supercoiling introduced by continuous elongation) and plot heatmaps of each of these variables calculated for weak (**row i**, $k_i = 0.008\,\mathrm{s}^{-1}$), moderate (**row ii**, $k_i = 0.05\,\mathrm{s}^{-1}$), and strong (**row iii**, $k_i = 0.02\,\mathrm{s}^{-1}$) promoters. We also fix either $k_G = k_G^*$ (**row iv**) or $k_T = k_T^*$ (**row v**) and vary the other topoisomerase activity (i.e., we trace along the blue lines in rows i-iii) and plot each aforementioned statistic.
(EPS)

**S4 Fig. Heatmaps on Regulatory Effect of Topological Domain Size.** The log2 fold changes in the transcription rate between pairs of simulations (red: higher; blue: lower) for weak **(column i)**, moderate **(column ii)**, and strong **(column iii)** promoters were plotted for varied TopoI and gyrase activities ($k_G$ from 0 to $2k_G^*$ and $k_T$ from 0 to $2k_T^*$). We plot the log2 fold change in the expression rate in response to **(A)** reducing the barrier distances from 10 kb to 1 kb, **(B)** reducing nonspecific topoisomerase activities 10-fold, **(C)** reducing the barrier distances from 10 kb to 1 kb and increasing topoisomerase nonspecific activities 10-fold to compensate, **(D)** increasing the barrier distances from 10 kb to 100 kb, **(E)** increasing the nonspecific topoisomerase activities 10-fold, and **(F)** increasing the barrier distances from 10 kb to 100 kb and decreasing the nonspecific topoisomerase activities 10-fold to compensate.
(EPS)

**S5 Fig. Simulated Fano Factors.** Fano factors for simulations ran for 1 kb genes with varied promoter strengths (**row i**, weak promoter, $k_i = 0.008\,\text{s}^{-1}$; **row ii**, moderate promoter, $k_i = 0.05\,\text{s}^{-1}$; **row iii**, weak promoter, $k_i = 0.2\,\text{s}^{-1}$), barrier distances (**column A**, $D = 1$ kb; **column B**, $D = 10$ kb; **column C**, $D = 100$ kb), and topoisomerase activities ($k_G$ from 0 to $2k_G^*$ and $k_T$ from 0 to $2k_T^*$, where $k_G^*$ and $k_T^*$ are the topoisomerase activities calculated to remove the supercoiling introduced by elongation at each barrier distance). Blue lines indicate where $k_G = k_G^*$ and $k_T = k_T^*$. Fano factors less than 1 are shown as 1. Such cases result from a forced spacing of initiation events due to bound RNAPs blocking the promoter. Fano factors are displayed on a logarithmic scale.
(EPS)

**S6 Fig. Response of Transcription to Gyrase or TopoI Inhibition.** We consider 1 kb genes flanked by 10 kb barriers (**columns A-B**) or 100 kb barriers **(columns C-D)** for varied promoter strengths (**row i**, weak promoter, $k_i = 0.008\,\text{s}^{-1}$; **row ii**, moderate promoter, $k_i = 0.05\,\text{s}^{-1}$; **row iii**, weak promoter, $k_i = 0.2^{-1}$) and topoisomerase conditions ($k_G$ varied from 0 to $2k_G^*$ and $k_T$ varied from 0 to $2k_T^*$, where $k_G^*$ and $k_T^*$ are the topoisomerase activities calculated to remove the supercoiling introduced by elongation at each barrier distance). We consider a 5-fold reduction in either $k_G$ **(columns A, C)** or $k_T$ **(columns B, D)** and plot the resulting log2-fold-change in the transcription rate.
(EPS)

**S7 Fig.** Effects of variations to the upstream or downstream barrier separately. (A) We reproduce the data of Boulas et al. Fig 2. [Boulas I, Bruno L, Rimsky S, Espeli O, Junier I, et al. (2023) Assessing in vivo the impact of gene context on transcription through DNA supercoiling. Nucleic Acids Res 51: 9509–9521. 10.1093/nar/gkad688.] with WebPlotDigitizer by automeris.io, showing the relative strength of various promoters alongside the change to expression strength on lengthening the upstream barrier distance to 3200 (Ai) or downstream barrier distance to 3408 bp (Aii) reported as the Far/Close ratio (i.e., the ratio of the expression strength and far and close distances, equivalent to the upstream and downstream susceptibilities discussed by Boulas et al.). We then consider simulations mimicking these conditions in silico by varying the upstream and downstream barrier distance independently. We consider a 1 kb gene located 250 bp from an upstream barrier and 320 bp from a downstream barrier. We then increase the upstream distance to 3200 bp **(column B)**, or we increase the downstream barrier distance to 3408 bp **(column C)**. We plot the resulting log2-fold-change for varied promoter strengths (**row i**, weak promoter, $k_i = 0.008\,\text{s}^{-1}$; **row ii**, moderate promoter, $k_i = 0.05\,\text{s}^{-1}$; **row iii**, weak promoter, $k_i = 0.2\,\text{s}^{-1}$) and topoisomerase activities ($k_G$ from 0 to $2k_G^*$ and $k_T$ from 0 to $2k_T^*$, where $k_G^*$ and $k_T^*$ are the topoisomerase activities calculated to remove the supercoiling introduced by elongation at each barrier distance). Blue lines indicate where $k_G = k_G^*$ and $k_T = k_T^*$. (D) In conditions of low TopoI activity ($k_T = 0.2k_T^*$) and high gyrase activity ($k_G = 2k_G^*$), we report the same Far/Close ratio for our simulations when we increase the upstream barrier distance from 250 bp to values ranging from 250 bp to 3200 bp (Di) and when we increase the downstream barrier distance from 320 bp to values ranging from 320 bp to 3408 bp (Dii), using the same promoter strengths as in (B-C). Cartoons created with BioRender.com.
(EPS)

## Author contributions

**Conceptualization:** Boaz Goldberg, Nicolás Yehya, Jie Xiao, Sam Meyer.

**Data curation:** Boaz Goldberg, Sam Meyer.

**Formal analysis:** Boaz Goldberg, Sam Meyer.

**Funding acquisition:** Jie Xiao, Sam Meyer.

**Investigation:** Boaz Goldberg, Jie Xiao, Sam Meyer.

**Methodology:** Boaz Goldberg, Nicolás Yehya, Jie Xiao, Sam Meyer.

**Project administration:** Jie Xiao, Sam Meyer.

**Software:** Boaz Goldberg.

**Supervision:** Jie Xiao, Sam Meyer.

**Visualization:** Boaz Goldberg, Sam Meyer.

**Writing – original draft:** Boaz Goldberg, Sam Meyer.

**Writing – review & editing:** Boaz Goldberg, Nicolás Yehya, Jie Xiao, Sam Meyer.

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
