## [Decision Letter · Decision Letter 0]

2 Mar 2025

PCOMPBIOL-D-25-00003

Differential effect of supercoiling on bacterial transcription in topological domains

PLOS Computational Biology

Dear Dr. Meyer,

Thank you for submitting your manuscript to PLOS Computational Biology. After careful consideration, we feel that it has merit but does not fully meet PLOS Computational Biology's publication criteria as it currently stands. Therefore, we invite you to submit a revised version of the manuscript that addresses the points raised during the review process.

Please submit your revised manuscript within 60 days May 02 2025 11:59PM. If you will need more time than this to complete your revisions, please reply to this message or contact the journal office at ploscompbiol@plos.org. Please include the following items when submitting your revised manuscript:

We look forward to receiving your revised manuscript.

Kind regards,

Mark Alber, Ph.D.

Section Editor

PLOS Computational Biology

Mark Alber

Section Editor

PLOS Computational Biology

**Journal Requirements:**

3) We notice that your supplementary Figures, and Table are included in the manuscript file. Please remove them and upload them with the file type 'Supporting Information'. Please ensure that each Supporting Information file has a legend listed in the manuscript after the references list.

Potential Copyright Issues:

i) Figures 1A, and 1C. Please confirm whether you drew the images / clip-art within the figure panels by hand. If you did not draw the images, please provide (a) a link to the source of the images or icons and their license / terms of use; or (b) written permission from the copyright holder to publish the images or icons under our CC BY 4.0 license. Alternatively, you may replace the images with open source alternatives. See these open source resources you may use to replace images / clip-art:

5)  Thank you for stating that "All data files are available from the Zenodo database at DOI: 10.5281/zenodo.14586107." This link reaches a DOI Not Found page. Please amend this to a working link or provide further details to locate the data.

**Reviewers' comments:**

Reviewer's Responses to Questions

Reviewer #1: The authors present a stochastic, Gillespie simulation based approach for simulating the differential effects of positive and negative supercoiling on bacterial transcription in confined topological domains. The simulation results are intriguing and allow a computational framework to explore how fluctuations in gyrase and topoisomerase concentrations cause polymerase stalling. The paper is well written, well motivated and deserves consideration for publication. Overall, I recommend one cycle of revision and re-review.

My primary critique of the paper, which is mostly minor, is that the modeling framework used in this paper is referred to as the Gillespie algorithm (or the SSA), but none of the conventions for presenting a Gillespie stochastic simulation are followed. It is standard to draw the chemical reaction network as a figure, to list the corresponding chemical reactions, and then to write down a table of the propensity functions associated with each reaction channel.

Simulation results and parametric tables are usually plotted (this part is done correctly in the paper). None of these conventions are observed which makes it very difficult for a reviewer to evaluate technical correctness, short of a full exploration and reverse engineering of the code. The notation in the code needs to reflect the notation used in the paper (which is basically absent since no presentation of the equations or reactions are made), but this is a minor omission that can be easily fixed.

Qualitatively, the results are very interesting. I especially find the stalling characteristic of the simulation framework interesting. It is worth noting that there are earlier works modeling the supercoiling-dependent nature of transcription initiation and elongation, especially as they related to gene expression in cell-free and in vivo bacteria (see the work of the Murray group by Yeung et al).

I was a bit confused as to why the authors used the Fano factor (a measure of deviation from a Poisson distribution) rather than the coefficient of variation to measure noisiness. It's clear the authors were trying to draw a connection with the literature, but it seems like the COV would be a scale-free metric for the amount of noise or burstiness in transcription.

Lastly, it seems the top-down ('omics") approach to validating what is a reductionist model of single-gene transcription seems a bit mismatched. This part of the paper seemed underdeveloped, so I would suggest thatthe authors exclude the section all together or expand on that section so that the reasoning is much more clear. Further, the number of parameters in the model seem far more than what the data can constrain, so it seems like it would be hard to justify the observations in nature as uniquely explained by the predictions from the model. Other parameterizations of the same model might yield distinct, contradictory predictions - this does not show the model is explanatory nor non-explanatory since we don't have a great sense of the perfect parameterization of the model. The core issue is model identifiability with such a generic collection of datasets from the literature. I don't have a solution for this challenge and while I find the comparison interesting, this may be a point of scrutiny/criticism from the scientific public.

Overall, the paper is very promising and I think it will be a great fit for PLOS CompBio. Please make the major revisions suggested and I would be happy to review it again.

Reviewer #2: Goldberg et al. set out to model DNA supercoiling and examine the effects of two types of topoisomerases: topoisomerase I and gyrase. They specifically modeled the recruitment of these enzymes to supercoiling sites as a function of supercoiling density and sign, resulting in complex, non-monotonic dynamics. The focus of their study was on the activity of a single gene, where they varied the strength of the promoter and the size of the domain.

The authors demonstrated that there is an optimal level of supercoiling—specifically negative supercoiling—for maximizing transcription rates. They found that weak promoters are not affected by slow elongation speeds but are limited by initiation rates. In contrast, strong promoters are constrained by RNA polymerase stalling. These differences lead to distinct sensitivities to topoisomerase I and gyrase for various promoters. The data suggest that there is an optimal level of activity for topoisomerase I and gyrase to achieve the highest level of gene activity, which may vary depending on the initiation rates of the promoters, essentially defining their strength.

Regarding domain size, smaller regions result in fewer relaxation events and more stalling. As a result, strong promoters in smaller domains are limited by topoisomerase activity. Conversely, larger domains favor transcription with lower topoisomerase I levels and higher gyrase levels. Interestingly, the authors show that strong promoters exhibit low noise (Fano factor below 1), indicating more coordinated bursting due to synchronization of RNA polymerase binding times. Finally, the authors examined multiple bacterial species to identify trends in expression patterns upon topoisomerase and gyrase inhibition, highlighting differences between strong and weak promoters.

Main points

1. There should be schematics to explain the models for the main figures. Figure four could especially benefit from cartoons illustrating what each row is trying to represent with respect to changes in the model. This will help with interpretation of the data. Similarly figure 3 could benefit from some cartoon graphics helping the reader to appreciate what is happening in the panels of data. Potentially both what is modeled and what is observed. Figure two could have a cartoon above the left panel of what the input distributions are schematically (e.g. delta function spikes at specific topoisomers, or a normal distribution).

2. Heatmaps are not the most effective way to present data. The text for the results convincingly supports its arguments, but Figures 3 and 4 obscure these arguments. For example, the authors argue in Figure 3 that there is an optimal amount of Top1 activity for a given promoter strength. This is visible in the heatmaps, but would be clearer if, for a few choices of gyrase activity, plot the transcription rate (or elongation speed, etc) as line charts. Effectively, take a cross-section of the heatmaps in order to clearly show non-monotonic behavior.

Another possibility is to make graphs that look like the data presented in FIgure 6. A more natural way to understand the Figure 3 results would be a line plot with “promoter strength” on the x axis and “normalized transcription"/other variables on the y axis, with the use of different colors or linestyles to separate lines. All in all, being able to map a specific text claim to a specific, explicative plot would be great!

3. Do similar results in Figure 4 hold if the domain size is held constant at 10kb, but the topoisomerase rate is 0.1x or 10x? The authors argue that most of the domain size trends are because of the relative activity of the topoisomerases. Discussing this case would strengthen that argument. Relatedly, in Figure 4, why are the y-axis tick marks the same in rows B and D, in the cases with 10x and 0.1x topo activity?

3. The assumptions regarding polymerase drag should be explicitly stated. The assumption of constant linking number upon elongation (e.g. the only change in linking number constraints is a result of gyrase/Top1 activity) assumes that the polymerase is so drag-limited that only the DNA, not the polymerase, rotates. The authors recognize this assumption when they say “the frictional drag experienced by RNAP might be weaker due to the absence of translation and a less crowded environment compared to the model assumption.”, but it would greatly clarify the model development part if this assumption is made more explicit.

4. Plasmid modeled as circular? Highly distant boundaries are not likely to replicate the predictions of a circled plasmid outside of the single-transcriptional-unit case presented. Even in the single-gene case, stochastic motion of polymerases should cause a small but non-zero amount of supercoiling due to diffusion around the backbone. This likely does not affect the presented results, but the authors should potentially call out this limitation. The more common way to encode circular boundary conditions with a linear coordinate is to “reflect” the first and last polymerase linking number constraints past the edges of the domain as “ghost” linking number constraints.

5. 2. The authors should explicitly state that “we assume that the linking number of polymerases does not change upon elongation; it only changes upon gyrase/topo level. Because linking number is excess twist / distance, constant linking number upon elongation means that all of the rotation is going into supercoiling generation. In other words, this assumes that the polymerase is so drag limited that it does not rotate”. This explains the Figure 2 statement that “maybe certain regimes are not this drag limited”

Minor points

1. Check the references for number 51 line 777. I'm not sure this is the intended reference

2. In the results section for Figure 5, the authors should discuss that the Fano factor is not scale-free, e.g. to get meaningful Fano factors, you need to know the exact number of transcripts, which requires either simulations or some method that counts discrete number of transcripts. This could motivate the choice of experimental data that the authors used.

3. In figure 5A/5C, potentially consider other, clearer plot types. Instead of an overplotted scatter plot in 5A, you could use faceted KDE plots to see how the overall distirbution shape changes. In 5C, overlapping barcharts obscure the marginal distribution changes. Consider a 1D KDE or a line-only histogram, so readers can clearly compare the Poisson distribution against the calculated distributions.

Reviewer #3: Reviewed by Ivan Junier.

Several biophysical models have been proposed over the years to better understand how the topological constraints associated with transcription (and also replication) are managed in the cell and what their impact is on gene expression. The work of Goldberg et al. follows this line and aims to understand the implications of certain models that were recently developed by the authors (ref [23]) or by research groups like ours (ref [17]). In particular, these models have demonstrated their ability to quantitatively rationalize complex expression data where physical constraints are varied in the case of an isolated gene [17], which is exactly the experimental situation discussed in this work.

In this context, the proposed manuscript seems to me to be an interesting contribution to the understanding of these models. However, for this work to be publishable, important changes and additional analyses are required.

First, the introduction of the paper is relatively vague regarding the range of models that have been proposed in the literature. A better contextualization is necessary. Otherwise, the authors’ model will appear as just another model. More specifically:

- A brief historical overview of the models seems necessary.

- It is essential to explain the similarities and differences between the proposed model and the most recent ones [17] and [23]. With respect to [17], I see two major differences: i) the consideration of the processivity of gyrase activity, which effectively generalizes [17], and ii) the absence of specific activity for topoisomerases — upstream of the gene for TopoI and downstream for DNA gyrase, which thus reduces the generalization power compared to [17]. In [23], authors considered the additional possibility for the RNAP to rotate.

- The authors mention that one of the advances of their model is the consideration of the continuous response to SC of topoisomerase activity. However, in lines 167-169, the authors mention that their work is not sensitive to the discrete or continuous approximations of topoisomerase activities. I therefore conclude that there is no need to consider a continuous version that requires more parameters.

- The authors mention that they take into account the issue of gyrase topoisomerase binding rates (also in the abstract). However, their model does not separate the binding mechanism from the activity once bound. Therefore, the authors should explain this better, particularly that their model, like others before it, considers an effective approximation of these two process.

Secondly, the authors justify the quantitative nature of their model by emphasizing its agreement with available experimental data. For instance, compared to Chong et al. [28], they also observe Fano factors around and above 2. However, I note that their distribution of transcript numbers is actually qualitatively different from that observed in [28]. In particular, as shown in Fig. 7B and Table S1 of [28], Chong et al. found a distribution of mRNAs with a very high number of cells showing no transcripts, which mathematically corresponds to a Poisson with zero spike distribution [28].

More importantly, an important comparison is missing in the proposed manuscript: that with experimental results obtained in [17] where a systematic study of the effect of barrier distances (downstrean and upstream) is performed, i.e., just as the framework the authors consider. In particular, the authors should investigate the conditions to observe an absence of an effect of downstream length on the expression rate of an isolated gene (Figure 2A in [17]).

Finally, the authors mention that their work “provides a unifying framework” (abstract and discussion). I do not agree. Indeed, in [17], it is shown that a quantitative explanation of the dependence on the size of the upstream domain requires considering the specific activity of TopoI, an operational mode that is not accounted for in the authors’ model.

In summary, I believe that the terms “quantitative” (lines 151 and 722) and “unification” (lines 60-61, 722) are not applicable here and that the authors should more clearly indicate the approximaitons and hypotheses they consider, as well as the differences with the experimental studies they cite. They also should perform additional analyses to compare their results with the experimental results in [17].

Additional comments (more or less important):

- The authors often use the term “regulation” where “impact/effect” would be more appropriate. For instance, “Regulation by topological domain size” (line 507) suggets that cells can regulate this aspect, which is discutable. The authors should revise the text accordingly.

- The first sentence of the articles says “SC has been proposed as a new means to regulate transcription independent of regulatory protein binding [1]”. I see two issues here. First, I do not think that [1] proposes this. Second, this is not new (as explained in [1]).

- Line 91 : I would add a work where they experimentally demonstrate this effect, e.g. the work by Leng et al. 10.1073/pnas.1109854108.

- Line 96-97 : I do not understand why [3] is cited here. I would cite at least Menzel and Gellert (1983) 10.1016/0092-8674(83)90140-x

- Line 101: related to te citation [8]: more recent reviews on topoisomerases exist and bring novel information not discussed in [8]. See for instance 10.1002/bies.202000286 and 10.1042/BST20240089

- Line 107 : “While SC may serve as a new, important transcriptional regulator” : see above about the term “new”

- Line 135-136 (and before): It is somewhat curious that [17] is not mentioned when introducing the single-gene approach. To my knowledge, [17] is the only experimental work that studies the impact of domain size on the expression rate of a single gene, exactly what is discussed in the proposed ms.

- Line 158: “Our model broadly follows an approach used in several previous works including ours [12, 17, 23-25]”. I see two problemes here: the lack of clear presentation of previous models (see general comments) and the mixing of models of different nature. For instance, the model used by the authors is directly related to [17] and [23]. The work [12] is of different nature as it does not consider stalling.

- Line 163: if the authors want to use the terminology “improvements”, they should also consider “deteriorations”. Namely, with respect to [17], i) they do not consider different stages for the initiation, which have been shown in [17] to be necessary to explain variations between promoters senstivty to the upstream distances, ii) they do not consider the specific activities of topoisomerases (see above). With respect to [23], they do not consider the pssibility for the RNAP to rotate.

- Line 166: the work by Ashley et al. could be added (10.1093/nar/gkx649)

- Line 177: a reference for plectoneme hopping could be added

- Line 195: I would add that TopoI binds single-stranded DNA.

- Line 217: a reference for the footprinting length of an elongationg RNAP could be added

- Line 228: this maximal speed of RNAP is for minimal growing conditions. This should be added.

- Line 282: “TopoI” instead of “gyrase”

- Line 285: the choice of $\sigma_T=\sigma_i=-0.04$ is discutable (and, hence, should be discussed) for at least two reasons: i) there is experimental work suggesting $\sigma_T=-0.05$ (10.1074/jbc.275.11.8103) and ii) differences between $\sigma_T$ and $\sigma_i$ have been shown to lead to qualiative different behaviors depending of the exact value of $\sigma_i$ [17]

- Line 227 and page 18: it looks that the simulations require to introduce a (sufficiently negative) $\sigma_{start}$. What is the in vivo meaning of this value, knowing that by coherence, $\sigma_{start}$ should be given by the activity of topoisomerases (and hence, should not be necessary)? This should be discussed.

- Line 296: I do not see how the authors conclude the value of $\rho_G=4$ from [36]. This should be explained. More importantly, the experimental work of Ashley et al. (10.1093/nar/gkx649) suggests a scenario in which gyrase activity involves a very high processivity intersperced with long periods of pauses, which differs from the scenario proposed by the authors. This should be discussed.

- Results of Fig. 2: this suggests a scenario different from what has been proposed in [13]. This could be discussed.

- Line 507: “Regulation by topological domain size” => “Impact of …” (see above)

- lines 508-510: this is really curious not to cite [17], here. Also, I do not see the relevancy of [48] here.

- lines 553-555: I note that there is nothing new here with respct to [17].

- lines 580-582: it is not clear why the authors mention this since the Fano factor is a classical quantity to assess variability in gene expression.

- line 620: “a moderate promoter (ki = 0.05 s-1) with intermediate barriers (D = 10 kb). This promoter strength and domain size (green triangles in Fig. 5A) are representative of many bacterial genes” : this is discutable. Indeed, it has been shown that transcription units, even those that are weakly expressed, often need to be considered within broader domains in which transcription is coordinated (when expressed) — see, for example, 10.1371/journal.pone.0155740. The idea that barriers 10 kb away from isolated genes (with in mind the size of topological domains in E. coli [43]) is a typical sitjuation is therefore hypothetical to me.

- In the scenario 2 (lines ~ 640), it should be explained how the gene goes back to the on-state.

- I note that the timescale of the off-state in scenario 2 is on the order of 10 minutes, in accord with experimental data (10.1038/ng.821). This could be mentionned.

- Line 665: in [17], there is no transcriptomics data

- lines 711 to 715: I do not understand why [50] supports the author’s statement. The work [50]concludes that there are approximately 300 gryases constantly bound along the chromosome, which is sufficient to relax supercoiling constraints in approximately the same number of supercoiled domains (see [43]). Meaning that there is in principle enough DNA gyrase ready to act on lowly expressed genes, contrary to what the authors state.

- Line 732: [17] does not show that “stronger genes are relatively repressed by TopoI inhibition”

- Line 766: “Boulas provides both experimental and computational evidence that transcription is mostly sensitive to the upstream barrier distance” : there is no computational evidence in [17]. The model is adjusted to fit the experimental evidence.

**Have the authors made all data and (if applicable) computational code underlying the findings in their manuscript fully available?**

Reviewer #1: Yes

Reviewer #2: Yes

Reviewer #3: **No: ** The code of the model to reproduce data is not provided.

PLOS authors have the option to publish the peer review history of their article (what does this mean? ). If published, this will include your full peer review and any attached files.

**Do you want your identity to be public for this peer review?** For information about this choice, including consent withdrawal, please see our Privacy Policy .

Reviewer #1: No

Reviewer #2: No

Reviewer #3: **Yes: ** Ivan Junier

**Figure resubmission:**

**Reproducibility:**



---

## [Decision Letter · Decision Letter 1]

22 Jul 2025

PCOMPBIOL-D-25-00003R1

Differential effect of supercoiling on bacterial transcription in topological domains

PLOS Computational Biology

Dear Dr. Meyer,

Thank you for submitting your manuscript to PLOS Computational Biology. After careful consideration, we feel that it has merit but does not fully meet PLOS Computational Biology's publication criteria as it currently stands. Therefore, we invite you to submit a revised version of the manuscript that addresses the points raised during the review process.

Please submit your revised manuscript within 30 days Sep 21 2025 11:59PM. If you will need more time than this to complete your revisions, please reply to this message or contact the journal office at ploscompbiol@plos.org. Please include the following items when submitting your revised manuscript:

We look forward to receiving your revised manuscript.

Kind regards,

Mark Alber, Ph.D.

Section Editor

PLOS Computational Biology

Mark Alber

Section Editor

PLOS Computational Biology

**Journal Requirements:**

1) We notice that your supplementary Tables are included in the manuscript file. Please remove them and upload them with the file type 'Supporting Information'. Please ensure that each Supporting Information file has a legend listed in the manuscript after the references list.

2) Some material included in your submission may be copyrighted. According to PLOSu2019s copyright policy, authors who use figures or other material (e.g., graphics, clipart, maps) from another author or copyright holder must demonstrate or obtain permission to publish this material under the Creative Commons Attribution 4.0 International (CC BY 4.0) License used by PLOS journals. Please closely review the details of PLOSu2019s copyright requirements here: PLOS Licenses and Copyright. If you need to request permissions from a copyright holder, you may use PLOS's Copyright Content Permission form.

Potential Copyright Issues:

i) Figures 4, and S7. We noted that you stated in the figures legends "Cartoons created with BioRender.com." Please confirm that you hold a Premium account and provide a pdf copy of the CC BY 4.0 Licence as provided by BioRender. For instructions on how to generate a CC BY 4.0 license for your figure, please see the guidelines here: https://help.biorender.com/hc/en-gb/articles/21282341238045-Publishing-in-open-access-resources. 

If you are using the free assets from BioRender, we are unable to publish these images as they are licenced under a stricter licence than CC BY 4.0. In this case we ask you to remove the BioRender images and replace them with open source alternatives.

See these open source resources you may use to replace images / clip-art:

- https://bioart.niaid.nih.gov/ 

- https://bioicons.com/

- https://healthicons.org/ 

- https://scidraw.io/

- https://reactome.org/icon-lib

- https://www.phylopic.org/images 

- https://journals.plos.org/plosbiology/article?id=10.1371/journal.pbio.3002395

**Reviewers' comments:**

Reviewer's Responses to Questions

Reviewer #1: The authors have adequately addressed all my concerns from the previous review cycle. I thank the authors for their responsiveness and willingness to consider my points.

I just seek clarification on one aspect. I am not sure what is meant by a deterministic-stochastic model. Is this a Gillespie simulation in which all the deterministic processes are stochastic processes with probability mass function of 1.0 (almost sure occurrence)? I found the discussion on lines 315-317 a bit vague still on this point. Also, the choice of the rate functions for the stochastic processes, specifically gyrase and topoisomerase binding seemed unclear.

In Figure 6C-6D the authors describe simulating multiple genes using the deterministic-stochastic model to see how topoisomerase inhibitors affect transcription levels. I am unclear as to how the authors accounted for the binding of transcription factors and other heterogeneous effects in simulating each gene. Is this is a generic "one gene" fits all model? Or is there something about each gene simulated that makes it actually represent a given gene in each of the bacterial species simulated across their transcriptome? This point seems critical to clarify.

After a minor revision addressing these points, I would be happy to review one last time for final publication.

Reviewer #2: The authors have substantially improved the manuscript and provided improved clarity of the figures. While there are still opportunities to improve the message of the figures, the accompanying text is sufficient for the reader to gain the main ideas. The message of figure 4 is particularly improved and the idea of changes in length and topo activity influencing one another is clear. The Fano number is an appropriate metric for these simulations. CV is nice but Fano is proabably better and more appropriate. The work is fine currently fine as is. We particularly enjoyed this line from the authors as often the appreciate between plasmid and chromosomes is not fully appreciated. "Within the current framework, the comparison suggests that topoisomerases affect transcription in a different manner in the plasmid employed in the latter study compared to most chromosomal genes analyzed in the transcriptomics data" Thanks for highlighting!

Below are minor suggestions for a bit more visual clarity.

- 1D could use some labels / sizing. The labels \sigma_s -\sigma_s are tiny for the black dashed lines.

- 1E could use some highlighting –

-Figure 2 Label the red line as -\sigma_s –

-Line 483: 3D? You mean Aiv? - Figure 4: scaling seems to be off.

-Fig 6A: needs an updated y-axis label "fold change _upon gyrase inhibition_"

- Figure 6: it would be nice to rearrange the overall figure, to have one row for all of the gyrase inhibition + the modeling result, then another row for all of the topisomerase inhibition + the modeling result.

-Line 449??? Somehow an emoji snuck in

Reviewer #3: The authors have provided a major revision that addresses all my comments and criticisms. I find the resulting manuscript clearer, more accurate, less emphatic, and more nuanced regarding the level of predictions this type of model can offer.

Overall, the study presents convincing new results, showing that models based on the physics of RNAP stalling—and how it is handled inside the cell—are becoming increasingly promising for understanding many aspects of gene expression. For this reason, I believe this work deserves publication.

More personally, and in response to the authors’ replies, I still feel that stating all models are “broadly” equivalent does not do justice to modeling efforts and tends to downplay the importance of model assumptions and their predictive power.

Minor comments:

- Lines 168-169: English issue

- Line 256: “Supercoiling” should be lowercase.

- Lines 261-267: This discussion directly concerns the code on which the authors have based their work (i.e., our model [21] based on RNAP stalling). As explained in [22], and as I have personally discussed with Sam Meyer, this code/model assumes that upstream and downstream torques act independently. This assumption differs from other models based on RNAP stalling. The impact of this choice on the results has never been studied and should be addressed in future work. Therefore, I find it somewhat awkward/misleading to refer to this aspect by stating that “some models have adopted a similar approach to model the stalling of elongating RNAP molecules [21]”.

- Line 449: Wrong symbol.

**Have the authors made all data and (if applicable) computational code underlying the findings in their manuscript fully available?**

Reviewer #1: Yes

Reviewer #2: Yes

Reviewer #3: Yes

PLOS authors have the option to publish the peer review history of their article (what does this mean? ). If published, this will include your full peer review and any attached files.

**Do you want your identity to be public for this peer review?** For information about this choice, including consent withdrawal, please see our Privacy Policy .

Reviewer #1: No

Reviewer #2: No

Reviewer #3: **Yes: ** Ivan Junier

**Figure resubmission:**
---

## [Decision Letter · Decision Letter 2]

24 Sep 2025

PCOMPBIOL-D-25-00003R2

Differential effect of supercoiling on bacterial transcription in topological domains

PLOS Computational Biology

Dear Dr. Meyer,

Thank you for submitting your manuscript to PLOS Computational Biology. After careful consideration, we feel that it has merit but does not fully meet PLOS Computational Biology's publication criteria as it currently stands. Therefore, we invite you to submit a revised version of the manuscript that addresses the points raised during the review process.

Please submit your revised manuscript within 60 days Nov 24 2025 11:59PM. If you will need more time than this to complete your revisions, please reply to this message or contact the journal office at ploscompbiol@plos.org. Please include the following items when submitting your revised manuscript:

We look forward to receiving your revised manuscript.

Kind regards,

Mark Alber, Ph.D.

Section Editor

PLOS Computational Biology

**Additional Editor Comments:**

Reviewer #1:

Reviewer #2:

Reviewer #3:

**Journal Requirements:**

**Reviewers' comments:**

Reviewer's Responses to Questions

**Comments to the Authors:**

Reviewer #1: This constitutes a final, editorial review for Boaz, Nicolas, Jie, and Sam. I find the paper to be very compelling and the results to be extremely useful for thinking about transcription, DNA supercoiling, and topoisomerase-based regulation. The paper presents a mechanism to explain and control the size of transcriptional bursts and a means to capture the dual-side tapered effect of optimal supercoiling on transcription initiation and elongation, as well as topoisomerase activity. The paper presents a novel mathematical model, based on a combination of deterministic and stochastic reactions. I recommend this paper for eventual publication, but only after a major revision to address extensive editorial issues (grammar, typos, and expository gaps).

*In the abstract, update this sentence to express the following: "SC therefore regulates transcription in the absence of transcription factor binding, in the context of chromosomal topological domains and the actions of topoisomerases."

Too often, overly strong claims or sweeping generalizations are asserted that need to be softened to reflect prior work or prior findings.

*Line 48: Clarify domain size of what?

*Line 50: “In this work we developed a quantitative model to describe SC-coupled transcription.” (This is not novel, see Galloway, Yeung for two recent examples and there are older examples in the literature). The authors do a good job of identifying some gaps in the literature but then do not connect those gaps with what they say they will accomplish in the paper. Differentiate the work from what has been done.

*Line 51: Given the comments from the authors in response to my previous review, this sentence in the abstract should be “transcription of a single isolated gene with varying promoter strengths and in a topological domain of varying sizes” should say something to the effect of a “we formulate a generalized, single gene model, agnostic to the specific regulatory characteristics of various gene networks within single prokaryotic and eukaryotic cells, to examine the predictive power of a model that articulates the effects of DNA supercoiling, enzymatic topoisomerase activity, and “domain size” and find that the model is surprisingly accurate in predicting XYZ ….”

*Line 56: Several bacterial species should be rephrased with a statement about the diversity of phyla or taxonomic classification of the species - e.g., we examined both gram-positive and gram-negative, or we examined a range of soil borne, water-borne, and mammalian host associated microbes…

*What is meant by “a relevant parameter range”?

*What is the global relationship? Just state the type of relationship in the abstract.

*A period is missing after the first sentence of the paper.

*A period is missing after the second sentence of the paper.

*Line 85: SC-induced torque can stall what?

*Line 102: Comma splice (grammatically incorrect comma)

*Line 111: [57] also develops a mathematical framework for modeling bacterial transcription and SC, but it does not take a computational biophysics focus.

*Line 113: [57] explicitly modeled the effect of supercoiling on transcription in two stages, initiation and elongation with separate regulatory dynamics. The two stages were modeled as deterministic though, which is probably inaccurate and too coarse grained at the molecular level.

*Line 114: Enzymatic terms of gyrase and TopoI activity were included in [57] but they did not take into account the bell-curve response of catalytic enzymes around optimal supercoiling setpoints.

*Line 117: [57] explicitly shows comparisons of quantitative fits of experimental measurement data with predictions for a transcription-supercoiling model. The model is able to predict the response in cell-free systems quite well, but it was not able to account for the heterogeneity in response when considering transcription factors. The model also made structured assumptions about gyrase and topoisomerase’s response curves, which were hypothetical, at best, since the authors lacked gyrase and topoisomerase response curves for their plasmid studies.

*Line 126-127: Again, reference [57] is completely omitted when the entire point of the study was to understand how neighboring genes impacted each other in various orientations on plasmids. Additionally, [57] shows how neighboring gene orientation can be used to dramatically improve performance of transcription in the toggle switch circuit.

*Line 128: “To the best of our knowledge”

*Line 134: The introduction to the single gene approach its quite abrupt. I think there is a lot of context missing here as to what one means by a “single-gene approach”. There should be some brief discussion about how this is meant as an abstraction of most or all genes in the genome, as conveyed by the authors in the helpful replies to this reviewer. As is, the phrase is referenced as if it is a common technique/modeling approach in the field and I doubt that this is the case. Perhaps one could invoke the “world is resistor” analogy that electrical engineers use, where a dramatic simplification is applied to the excluded elements of the model, with a primary focus on one component, the gene of interest.

*The introduction of the modeling is still extremely vague. The authors should clearly state upfront what the model is, what type of model class it is, express the model equations for the deterministic components and the reaction kinetics for the stochastic components. Instead, the authors introduce the model by drawing comparisons to references in the literature, suggesting what it adds or what it is not. This is a terribly confusing way to introduce a mathematical model. For a sound example of how to introduce a mathematical model, refer to this paper by Sam Meyer and Chaoming Song (2010) or this paper by Sam Meyer and Jost (2010).

*Line 172-Line 183: This reads more like an appropriate introduction of the model but it still lacks detail and digresses into a discussion of literature and potential variations on the model and justifications for the current choice (where the reader is still confused as to what the current choice actually entails). We also get a stronger sense of the model content in the caption of Figure 1, but this exposition is not had in the main text of the paper. It should be in both.

*Line 254: What is the justification for using beta_i = 0.005? It doesn’t necessarily follow from what is stated in Line 253. Please be more careful and explicit in your logical reasoning.

*Line 257: The postulated 25 bp elongation rate could use a literature reference.

*Line 286 should have commentary about what you are not using the coefficient of variation, a scale-free metric for intrinsic, normalized variation of a random process/variable.

*Line 322 should state any potential package dependencies and provide potential links to python code. Why was the simulation split across two computing languages?

*What is the justification for what is simulated deterministically versus stochastically? This seems somewhat arbitrary since elongation and termination could be viewed as stochastic processes comparable to initiation and topoisomerase binding(which were simulated stochastically). Why is RNA decay simulated stochastically while SC diffusion is deterministic? Both seem like first order processes — what is the deciding criterion?

Line 640 etc: Define what scenario 1 and scenario 2 and scenario 3 actually entail rather than just referring to their existence and the corresponding outcomes.

*Line 785: The model is not fully stochastic. Please update to be accurate and consistent with the definition in the paper. It is a hybrid deterministic-stochastic model.

*Line 870: What is this line trying to say? This should be the final draft of the paper and it seems like the writing is still in rough draft form.

*Does the paper need a conclusion section?.

*Supplemental Figure 1: Should there be error bars?

*Supplemental Figure 2: What does the red time-series trace on the "Weak Promoter" simulation end prematurely?

This could be potentially annotated or overlayed to explain the significance of the premature end of the red trace.

All in all, I feel this paper should be published in PLOS Comp Bio after a major revision. I encourage the lead author to refine the quality and polish of the writing in this paper, to abide the highest of academic standards. I would be happy to review the paper one last time, if needed. Thank you for reading.

Reviewer #2: Congrats on the interesting work!

Reviewer #3: The new manuscript is ready for publication.

**Have the authors made all data and (if applicable) computational code underlying the findings in their manuscript fully available?**

Reviewer #1: Yes

Reviewer #2: Yes

Reviewer #3: Yes

PLOS authors have the option to publish the peer review history of their article (what does this mean? ). If published, this will include your full peer review and any attached files.

**Do you want your identity to be public for this peer review?** For information about this choice, including consent withdrawal, please see our Privacy Policy .

Reviewer #1: No

Reviewer #2: No

Reviewer #3: **Yes: ** Ivan Junier

**Figure resubmission:**

**Reproducibility:**



---

## [Editor Report · Decision Letter 3]

30 Oct 2025

PCOMPBIOL-D-25-00003R3

Differential effect of supercoiling on bacterial transcription in topological domains

PLOS Computational Biology

Dear Dr. Meyer,

Thank you for submitting your manuscript to PLOS Computational Biology. After careful consideration, we feel that it has merit but does not fully meet PLOS Computational Biology's publication criteria as it currently stands. Therefore, we invite you to submit a revised version of the manuscript that addresses the points raised during the review process.

Please submit your revised manuscript within 30 days Dec 30 2025 11:59PM. If you will need more time than this to complete your revisions, please reply to this message or contact the journal office at ploscompbiol@plos.org. Please include the following items when submitting your revised manuscript:

We look forward to receiving your revised manuscript.

Kind regards,

Mark Alber, Ph.D.

Section Editor

PLOS Computational Biology

Mark Alber

Section Editor

PLOS Computational Biology

**Journal Requirements:**

We ask that a manuscript source file is provided at Revision. Please upload your manuscript file as a .doc, .docx, .rtf or .tex. If you are providing a .tex file, please upload it under the item type u2018LaTeX Source Fileu2019 and leave your .pdf version as the item type u2018Manuscriptu2019.

**Reviewers' comments:**

**Figure resubmission:**
---

## [Editor Report · Decision Letter 4]

3 Nov 2025

Dear %TITLE% Meyer,

We are pleased to inform you that your manuscript 'Differential effect of supercoiling on bacterial transcription in topological domains' has been provisionally accepted for publication in PLOS Computational Biology.

Best regards,

Mark Alber, Ph.D.

Section Editor

PLOS Computational Biology

Mark Alber

Section Editor

PLOS Computational Biology

---

## [Editor Report · Acceptance letter]

PCOMPBIOL-D-25-00003R4

Differential effect of supercoiling on bacterial transcription in topological domains

Dear Dr Meyer,

I am pleased to inform you that your manuscript has been formally accepted for publication in PLOS Computational Biology. Your manuscript is now with our production department and you will be notified of the publication date in due course.

With kind regards,

Anita Estes
